# Sparsity May Cry: Let Us Fail (Current) Sparse Neural Networks Together!

**Shiwei Liu**[1]\*, **Tianlong Chen**[1]\*, **Zhenyu Zhang**[1], **Xuxi Chen**[1], **Tianjin Huang**[2],
**Ajay Jaiswal**[1], **Zhangyang Wang**[1]
[1]University of Texas at Austin   [2]Eindhoven University of Technology
`shiwei.liu@austin.utexas.edu; t.huang@tue.nl`
`{tianlong.chen,zhenyu.zhang,xxchen,ajayjaiswal,atlaswang}@utexas.edu`

## Abstract

Sparse Neural Networks (SNNs) have received voluminous attention predominantly due to growing computational and memory footprints of consistently exploding parameter count in large-scale models. Similar to their dense counterparts, recent SNNs generalize just as well and are equipped with numerous favorable benefits (e.g., low complexity, high scalability, and robustness), sometimes even better than the original dense networks. As research effort is focused on developing increasingly sophisticated sparse algorithms, it is startling that a *comprehensive benchmark to evaluate the effectiveness* of these algorithms has been highly overlooked. In absence of a carefully crafted evaluation benchmark, most if not all, sparse algorithms are evaluated against fairly simple and naive tasks (eg. CIFAR-10/100, ImageNet, GLUE, etc.), which can potentially *camouflage* many advantages as well unexpected predicaments of SNNs. In pursuit of a more general evaluation and unveiling the true potential of sparse algorithms, we introduce "**Sparsity May Cry**" **Benchmark (SMC-Bench)**, a collection of carefully-curated 4 diverse tasks with 10 datasets, that accounts for capturing a wide range of domain-specific and sophisticated knowledge. Our systemic evaluation of the most representative sparse algorithms reveals an important obscured observation: *the state-of-the-art magnitude- and/or gradient-based sparse algorithms seemingly fail to perform on SMC-Bench* when applied out-of-the-box, sometimes at significantly trivial sparsity as low as $5\%$. The observations seek the immediate attention of the sparsity research community to reconsider the highly proclaimed benefits of SNNs. We further conduct a thorough investigation into the reasons for the failure of common SNNs. Our analysis points out that such failure is intimately related to the "lazy regime" of large model training, which hints us with stronger pruning recipes that alleviate the failure on SMC-Bench (though still more or less suffering). By incorporating these well-thought and diverse tasks, SMC-Bench is designed to favor and encourage the development of more scalable and generalizable sparse algorithms. We open-source SMC-Bench to assist researchers in building next-generation sparse algorithms that scale and generalize: `https://github.com/VITA-Group/SMC-Bench`.

## 1 Introduction

Sparse Neural Networks (SNNs) are no stranger to the deep learning community (Liu & Wang, 2023), but recently they have received stupendous attention in the era of transformers (eg. BERT, GPT, ViT, CLIP, etc.), when the parameter count is frequently measured in billions rather than millions. Due to the consistent efforts of sparsity researchers, SNNs have ushered enormous breakthroughs and can generalize just as well as original dense networks, and it is feasible to procure them after training (Frankle & Carbin, 2019; Sanh et al., 2020; Chen et al., 2020; Frankle et al., 2020), during training (Zhu & Gupta, 2017; Gale et al., 2019; Liu et al., 2021b), and even before training (Mocanu et al., 2018; Lee et al., 2019; Liu et al., 2022) their dense counterparts using pruning. Apart from well-known efficiency benefits, surprisingly, SNNs also enjoy auxiliary benefits such as adversarial robustness (Guo et al., 2018; Özdenizci & Legenstein, 2021; Chen et al., 2022), out-of-distribution

---

\*These authors contributed equally to this work.

generalization (Zhang et al., 2021; Diffenderfer et al., 2021), and uncertainty estimation (Liu et al., 2021a), etc. Despite the multi-dimensional success of numerous sparse algorithms, startlingly, our extensive survey across over 100 recent SNN papers within 2015-2022 unveils multiple daunting issues regarding evaluation datasets and protocols blindly followed within the sparsity community, that may significantly impede future progress if left unacknowledged.

**Issues with current evaluation paradigm:** *Firstly*, the vast majority of current work on SNNs is *narrowly evaluated*, i.e., only targeting a single or a few tasks (usually on image classification and sometimes on language understanding) on which SNNs have already proven their proficiency (Gale et al., 2019; Frankle & Carbin, 2019). Surprisingly, 79 papers out of our carefully selected 100 papers on SNNs, evaluate sparse models *merely* on a single task, where 72 out of them evaluate image classification. *Secondly*, people are obsessed with evaluating SNNs on well-understood datasets, including but not limited to MNIST (LeCun, 1998) (26 papers), CIFAR-10/100 (Krizhevsky et al., 2009) (59 and 37 papers, respectively), ImageNet (Deng et al., 2009) (62 papers), and GLUE (Wang et al., 2018) (9 papers), where deep neural networks have already exceeded the human-equivalent performance (refer to Appendix D for more details). For instance, even though ImageNet has been considered a rather challenging task over years, very high accuracy ($>90\%$) has been reported many times (Yu et al., 2022; Wortsman et al., 2022; Zhai et al., 2022). Such relatively restricted evaluations with "nearly saturated performance" limit the scope of sparse neural networks and are potentially ill-suited to identify new and unexpected capabilities of SNNs.

Addressing the aforementioned limitations of current SNN evaluation protocols is a pressing need for the community. To this end, we assemble a large-scale, fairly arduous, and diverse benchmark for sparse neural networks - "**Sparsity May Cry**" **Benchmark** (or briefly **SMC-Bench**). Specifically, we consider a broad set of tasks including *complex reasoning, multilingual translation, and protein prediction*, whose content spans multiple domains. Those tasks require a vast amount of commonsense knowledge, solid mathematical and scientific backgrounds to solve even for humans. Note that none of the datasets in SMC-Bench was created from scratch for the benchmark, we rely on pre-existing datasets as they have been agreed by researchers as challenging, interesting, and of high practical value. We rigorously measure the performance of state-of-the-art (SOTA) pruning and sparse training approaches (in their most common, basic settings) on SMC-Bench, to understand the potential of SNNs to scale and generalize. Our key observations and contributions can be unfolded as:

- We present "**Sparsity May Cry**" Benchmark, to **re-define** the evaluation protocols for sparse neural networks and facilitate a comprehensive assessment of SOTA sparse algorithms. The premise of SMC-bench is to develop a suite of large-scale, challenging, realistic, and diverse tasks and datasets that can empower the rigorous advancements in the community.

- SMC-Bench unveils a critical and startling observation - all of the SOTA sparse algorithms seem to fail on SMC-Bench "out-of-the-box", sometimes at significantly trivial sparsity *e.g.,* $5\%$. Note that the failure does not appear specific to one sparsification approach but unanimously across all approaches we evaluated. This observation *alarmingly* demands the attention of the sparsity community to reconsider the highly proclaimed benefits of SNNs.

- We conduct extensive experiments across representative SNNs produced by various SOTA pruning and sparse training approaches on SMC-Bench, and we summarize our findings: ❶ Model prunability is intimately related to task difficulty: models trained on difficult tasks suffer more from pruning compared to easier tasks; ❷ The success of before-training sparsification (sparse training or pruning at initialization) is hard to generalize in more complex scenarios; ❸ Iterative magnitude pruning (IMP) does not necessarily generalize better than one-shot pruning (OMP) or during-training pruning; ❹ Despite performance difference, different magnitude-based pruning approaches lead to extremely similar layerwise sprasities.

- We further carry out a comprehensive investigation into the potential causes of SNN failures on SMC-Bench. Our analysis suggests that the failure of the existing sparse algorithms might be due to the "lazy regime" dynamics emerging in sufficiently overparameterized models (Chizat et al., 2019; Malladi et al., 2022). Inspired by this finding, we hypothesize and confirm that the second-order pruning approaches, i.e., oBERT (Kurtic et al., 2022), are more reliable pruning approaches for SMC-Bench, which yield relatively more promising performance on SMC-Bench in Appendix C.

## 2 RELATED WORK

### 2.1 ADVANCES IN SPARSE NEURAL NETWORKS

**Post-Training.** SNNs refer to a neural network where a certain portion of its components (e.g., weights, neurons, filters, and attention heads) have exactly zero values. The initial purpose of SNNs is retrospectively to accelerate model at inference time (a.k.a., post-training sparsification; Mozer & Smolensky (1989); LeCun et al. (1990)). Thanks to the over-parameterization property of deep neural networks, we can dramatically prune deep neural networks to smaller sizes with marginal loss of performance. Post-training sparsification has been well studied and results in various mature criteria that can be generally categorized into zero-order methods (magnitude-based; Han et al. (2015)), first-order methods (gradient-based; Molchanov et al. (2016); Sanh et al. (2020); Jiang et al. (2021)), and second-order methods (Hessian-based; LeCun et al. (1990); Hassibi & Stork (1992); Dong et al. (2017)). Second-order sparsification usually achieves higher performance than the other two but is also more expensive due to the full Hessian calculation. Fortunately, many approaches have been proposed to efficiently approximate Hessian (Zeng & Urtasun, 2018; Wang et al., 2019; Singh & Alistarh, 2020). The Lottery Ticket Hypothesis (LTH) adopts iterative magnitude pruning (IMP) on fully trained networks and finds a subnetwork at initialization that can be re-trained in isolation to match the original dense networks. Renda et al. (2020) further found that instead of re-training with the initial weights, re-training with the final weights achieves better performance. With the rise of large language models (LLMs), newer post-training pruning methods have emerged which aim to improve the affordability of these models (Sanh et al., 2020; Chen et al., 2020; Zafrir et al., 2021; Kurtic et al., 2022; Xu et al., 2021; Lagunas et al., 2021; Zhang et al., 2022; Frantar et al., 2021).

**During-Training.** During-training sparsification (Finnoff et al., 1993) is a cheaper option, compared to sparsify a fully converged model. Approaches of during-training sparsification usually train a dense network for some time and then gradually sparsify the network with some schedules and end up with a sparse model with target sparsities. Zhu & Gupta (2017); Gale et al. (2019); Lin et al. (2020); Liu et al. (2021b) are highlight approaches that gradually prune networks during training and meanwhile allow the pruned weights to be reactivated in case of inaccurate pruning. Another direction of during-training sparsification is adding sparsifying penalties such as (grouped) $L_0$ and $L_1$ norm to the loss function, which will punish the unimportant weights to zero, leading to sparse weights (Louizos et al., 2018; Luo & Wu, 2020; Savarese et al., 2020).

**Prior-Training.** Recently, foundation models (Brown et al., 2020; Chowdhery et al., 2022; Ramesh et al., 2022) have demonstrated promising quantitative improvement and new qualitative capabilities with increasing scale (Zhang et al., 2020b). Along with the scaling of model size and data size, the training resources of these foundation models also get outrageous. To accelerate training, we need to sparsify models before training. LTH unveils the possibility to find SNNs at initialization that can match their dense counterparts, even though it uses post-training pruning to find them. At the same time, sparse training (Mocanu et al., 2018; Mostafa & Wang, 2019; Dettmers & Zettlemoyer, 2019; Evci et al., 2020; Liu et al., 2021c; Schwarz et al., 2021) was proposed that can train a randomly-initialized sparse neural network from scratch while dynamically optimizing the sparse connectivity with promising performance. Instead of randomly initializing sparse networks, one iteration (Lee et al., 2019; Wang et al., 2020) or a few iterations (Tanaka et al., 2020; de Jorge et al., 2021) of training can be utilized to guide the search for sparse networks before training.

### 2.2 BENCHMARKING IN SPARSE NEURAL NETWORKS

Gale et al. (2019) rigorously evaluated variational dropout (Molchanov et al., 2017), $l_0$ regularizaion (Louizos et al., 2018), and GMP (Zhu & Gupta, 2017) on two large-scale tasks. They demonstrated that the appealing performance advantages of variational dropout and $l_0$ regularization cannot generalize to large-scale tasks whereas simple magnitude pruning performs surprisingly well. Liu et al. (2018) examined two pipelines: training from scratch and fine-tuning, concluding that fine-tuning a pruned model only gives comparable or worse performance than training from scratch. Blalock et al. (2020) provided a comprehensive literature review on SNNs and found that pruning papers rarely make direct and controlled comparisons. Frankle et al. (2021) assessed the efficacy of various pruning at initialization approaches and attribute their inferior performance to their insensitivity to weight shuffling and re-initialization. Liu et al. (2022) re-evaluated the performance of various random pruning before training and found that sparsely training a randomly pruned network from scratch can surprisingly match the performance of its dense equivalent. These papers shed light on the behavior of SNNs and discover important research problems for future work.

## 3    SMC-BENCH

SMC-Bench is crafted for evaluating if all proclaimed benefits of SNNs can "scale and generalize". It consists of 4 diverse and difficult tasks, including commonsense reasoning, arithmetic reasoning, multilingual translation, and protein prediction, with 10 datasets collected from prior work and open-source GitHub repositories. To investigate if there is a strong correlation between model prunability and task difficulty, we choose multiple datasets with different degrees of difficulty.

### 3.1    COMMONSENSE REASONING

Commonsense reasoning task asks commonsense questions about the world involving complex semantics that often require rich common sense and background knowledge. We consider three commonly used datasets for commonsense reasoning. (1) **RACE** (Lai et al., 2017) contains near 28,000 passages and 100,000 questions collected from the English exams for Chinese students in middle (RACE-M) and high school (RACE-H). (2) **WinoGrande** (Sakaguchi et al., 2021) is a modified version of the Winograd Schema Challenge (WSC) benchmark (Levesque et al., 2012) with enhanced scale and hardness, containing 44k problems. (3) **Commonsense QA (CSQA)** (Talmor et al., 2018) is a challenging dataset containing 12,247 multiple-choice questions where one source concept and three target concepts are first extracted from ConceptNet (Speer et al., 2017) based on which the Crowd-works are asked to author multiple-choice questions with two additional distractors. In general, CSQA is harder than WinoGrande and RACE, with ceiling human performance of 89%, 94%, and 95%, respectively.

### 3.2    ARITHMETIC REASONING

Arithmetic reasoning poses a question of a math problem and the model is asked to generate a mathematical equation that can be evaluated to get the answer. We consider the following three math word problem datasets: (1) the widely used **MAWPS** benchmark (Koncel-Kedziorski et al., 2016) composed of 2,373 problems; (2) the arithmetic subset of ASDiv (Miao et al., 2021) - **ASDiv-A** with 1,218 math problems; (3) the more challenging **SVAMP** (Patel et al., 2021) dataset which is created by applying complex types of variations to the samples from ASDiv-A. The task difficulty monotonically increases from MAWPS to ASDiv-A, and to SVAMP.

### 3.3    PROTEIN THERMOSTABILITY PREDICTION

Maintaining a stable 3D structure is an essential pre-condition for protein to function correctly in biological phenomena. Numerous efforts are devoted to modeling and predicting protein's stability against pH, salinity, and temperature. We consider the tasks of protein thermostability prediction on two representative datasets: (1) **HotProtein** (Chen et al., 2023) is recently proposed as a large-scale, standardized protein benchmark with organism-level temperature annotations, which contains 182K protein sequences and 3K folded structure from 230 different species. Three dataset variants, **HP-S**, **HP-S$^2$C**5, and **HP-S$^2$C**2, are adopted to examine sequence- and structure-based methods, respectively. **HP-S** has $\{6, 390, 3, 4946, 30, 333, 79, 087, 31, 549\}$ protein sequences from five categories of $\{Cryophilic, Psychrophilic, Mesophilic, Thermophilic, Hyperthermophilic\}$; **HP-S$^2$C**5 has both sequences and structures for $\{73, 387, 195, 196, 189\}$ proteins from the same five classes ordered from *Cryophilic* to *Hyperthermophilic*; **HP-S$^2$C**2 has both sequences and structures for $\{1, 026, 939\}$ proteins from $\{$"hot" $(\geq 45°\text{C})$, "cold" $(< 45°\text{C})\}$ two classes. (2) **Meltome Atlas** (Jarzab et al., 2020) is another challenging test bed for protein's thermostability. It has $\{7, 902, 15, 833, 10, 518\}$ protein sequences from three of the five aforementioned classes, from *Mesophilic* to *Hyperthermophilic*. All samples are annotated with their melting temperature.

### 3.4    MULTILINGUAL TRANSLATION

Multilingual translation processes multiple languages using a single language model and requires the model to have the ability to perform translation across languages. We follow Liu et al. (2020); Tang et al. (2020) and choose 10 English-centric language pairs (Fr, Cs, De, Gu, Ja, My, Ro, Ru, Vi, Zh $\leftrightarrow$ En) from an open source parallel corpus - OPUS (OPU, 2020). We follow Arivazhagan et al. (2019) and use pivot data through English to create 3 Many-to-Many multilingual translation fine-tuning settings including 2-to-2 (Fr, Cs), 5-to-5 (Fr, Cs, De, Gi, Ja), and 10-to-10.

## 4 EVALUATION ON SMC-BENCH

**Models.** Despite we are aware that performing few-shot prompting on large-scale pre-training language models with billions of parameters is able to solve these tasks (Wei et al., 2022; Srivastava et al., 2022), we choose to focus on fine-tuning or training with pre-trained mid-scale models with millions of parameters, to improve the accessibility of our Benchmark. Specifically, we choose to fine-tune the popular RoBERTa (Liu et al., 2019) for commonsense reasoning; to fine-tune mBART (Liu et al., 2020) for multilingual translation; to train GTS (Xie & Sun, 2019) and Graph2Tree (Zhang et al., 2020a) with RoBERTa's pre-trained embedding for arithmetic reasoning; to fine-tune Transformer-based (Vaswani et al., 2017) for protein property prediction. See Appendix A for full details.

**Sparse Neural Networks.** We select the most representative magnitude- and/or gradient-based approaches where the prune operation is performed before, during, or after training. Formally, given a dense network $\theta_l \in \mathbb{R}^{d_l}$ with a dimension of $d_l$ in each layer $l \in \{1, \ldots, L\}$, pruning generates binary masks $m_l \in \{0, 1\}^{d_l}$ yielding sparse neural networks with sparse weights $\theta_l \odot m_l$. The sparsity level is the fraction of the weights that are zero-valued, calculated as $s = 1 - \frac{\sum_l m_l}{\sum_l d_l}$. Following a mainstream convention in many sparse training papers (Frankle & Carbin, 2019; Gale et al., 2019; Evci et al., 2020; Lee et al., 2019; Liu et al., 2021c), we sparsify most layers in the model including embedding layers and classifier heads, and we do not apply advanced techniques such as Iterative Learning Rate Rewinding (Renda et al., 2020) and Knowledge Distillation (Hinton et al., 2015) in our main evaluations, even if we observe that they help to alleviate accuracy drops as in Appendix C.

• *Lottery Ticket Hypothesis (LTH)* (Frankle & Carbin, 2019) is a strong post-training pruning baseline that iteratively adopts magnitude pruning after training to produce binary masks and re-train together with weights from step $t$. We set $t = 0$ in this paper, since rewinding to early training does not notably improve the performance of Transformer models (e.g., BERT) for downstream tasks (Chen et al., 2020). The pruning rate of each IMP is set as 20%.

• *Magnitude After Training* is a strong baseline for prune after training, which has demonstrated strong results in various regimes. After training or fine-tuning models on the specific task, we prune the model with one-shot magnitude pruning and re-train it with the full learning rate schedule from the beginning, dubbed "OMP (After)" in our experiments.

• *Random After Training* (Mittal et al., 2019) is the most naive baseline for post-training pruning. It uniformly samples a random score $s_l \in \text{Uniform}(0, 1)$ for each weight and prunes the weights with the lowest scores. After pruning, we also re-train with the entire training schedule.

• *Gradual Magnitude Pruning (GMP)* (Zhu & Gupta, 2017; Gale et al., 2019) gradually sparsifies networks during training according to a pre-defined sparsification schedule with sorting-based weight thresholding. The starting and the ending iteration of the gradual sparsification process are set as 10% and 80% of the entire training iterations. The frequency of sparsification steps is tuned among 500, 1000, and 4000, depending on the specific tasks. While we are aware of the advanced gradual pruning methods - movement pruning (Sanh et al., 2020), it usually exceeds GMP only at high sparsities (e.g., >90%), which is interesting but not within the scope of this paper.

• *Magnitude Before Training* (Frankle et al., 2021) simply removes weights with the lowest magnitude at initialization. Since we inherit weights from pre-trained models, the initial weights actually refer to the weights that are learned on the pre-trained tasks. We abbreviate this approach to "OMP (Before)" as we use one-shot magnitude pruning.

• *Random Before Training* (Liu et al., 2022) is the most naive baseline for prior-training pruning. We randomly sample scores for each weight and removes the weights with the lowest scores. Different from Random After Training, the pruning operation is performed before fine-tuning.

• *SNIP* (Lee et al., 2019) is a prior-training pruning technique that globally removes weights with the lowest connection sensitivity score $|g \odot w|$. SNIP is a strong baseline that consistently performs well among various prior-training approaches (Frankle et al., 2021).

• *Rigging the Lottery (RigL)* (Evci et al., 2020) is a leading sparse training method that updates the topology of sparse neural networks during training via a prune-and-grow scheme. To evaluate its effectiveness on downstream fine-tuning, we combine RigL with the other three prior-training methods. The update interval of RigL is set the same as the ones used for updating sparsity in GMP, following Liu et al. (2021b).

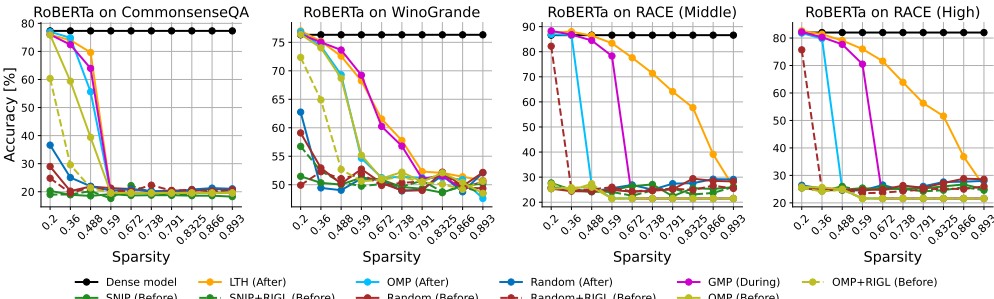

Figure 1: Commonsense reasoning performance of various sparse RoBERTa.

## 4.1 COMMONSENSE REASONING

**Implementation Details.** We follow the training settings of sequence modeling toolkit Fairseq (Ott et al., 2019) and fine-tune the pre-trained RoBERTa on our datasets with a standard cross-entropy loss. Specifically for each question, we also construct five inputs, one for each of the five candidate answer choices. Each input is constructed by concatenating the question and candidate answer together. We then encode each input and pass the resulting "[CLS]" representations through a classifier to predict the correct answer. All models are trained with the Adam (Kingma & Ba, 2014) optimizer with a learning rate of $1 \times 10^{-5}$ using an A100 GPU. For CSQA, we train the model for 3000 steps with a linear warmup of 150 steps and a batch size of 16. The dropout rate is set as 0.1. This gives us a test accuracy of 77.3% with dense RoBERTa. For RACE, we train each model for 3 epochs with a batch size of 16. This gives us 86.6% and 82.0% dense accuracy on RACE (H) and RACE (M), matching the ones reported in Fairseq (86.5% and 81.3%). Models on WinoGrande are trained for 23, 750 steps with 2, 735 warmup steps and 32 batch size, reaching a 76.3% accuracy.

**Results and Analyses.** The results of various sparse neural networks are demonstrated in Figure 1. We summarize our main observations below:

① *All sparse algorithms seemingly fail to find matching SNNs, even at trivial sparsities such as 36%.* While several methods maintain the dense performance at 20% sparsity, their performance starts to drop significantly after that, and will undergo a catastrophic failure as the sparsity continues increasing. It is difficult even for the top-performing LTH to maintain the matching performance after the $2^{rd}$ IMP iteration. This is in stark contrast with the behavior of SNNs on the image classification task, where LTH can gracefully preserve the matching performance even at very extreme sparsities (>95% on CIFAR-10/100 (Yin et al., 2022) and >80% on ImageNet (Renda et al., 2020)).

② *The quality of SNNs on harder tasks suffers more from sparsity.* Models trained on the hardest task, CSQA, undergo a larger accuracy loss at the same sparsity than the other two datasets. For instance, all the SNNs on CSQA suffer from a catastrophic accuracy drop (up to 74%) and become no better than the random prediction at 59% sparsity. Meanwhile, when trained on WinoGrande and RACE at 59% sparsity, two sparse algorithms (LTH and GMP) can maintain relatively good performance with a smaller performance loss (i.e., 3% ∼ 10%).

③ *Post-training pruning consistently outperforms prior-training pruning.* LTH achieves the best performance across datasets, GMP performs well, and OMP (After) follows behind. However, prior-training pruning achieves worse performance. OMP (Before) performs closely behind OMP (After), whereas SNIP performs no better than the naive random pruning. After digging deeper into the case of SNIP, we find SNIP aggressively prunes the embedding layers to more than 99% sparsity even with a mild overall sparsity of 20%. Surprisingly, the leading dynamic sparsity approach RigL does not bring significant gains to these prior-training approaches, and sometimes even hurts the performance.

## 4.2 ARITHMETIC REASONING

**Implementation Details.** We follow SVAMP (Patel et al., 2021) and choose the two top-performing tree-based models for arithmetic reasoning: GTS (Xie & Sun, 2019) and Graph2Tree (Zhang et al., 2020a). Graph2Tree in general performs slightly better than GTS. GTS adopts LSTM to encode input sequences and a tree-based Decoder to generate questions. Graph2Tree uses a graph transformer to learn the latent quantity representations from data, and a tree structure decoder to generate a solution expression tree. We follow exactly the training settings of Patel et al. (2021). The embedding weights are inherited from THE pre-trained RoBERTa. All models are trained with Adam for 50 epochs.

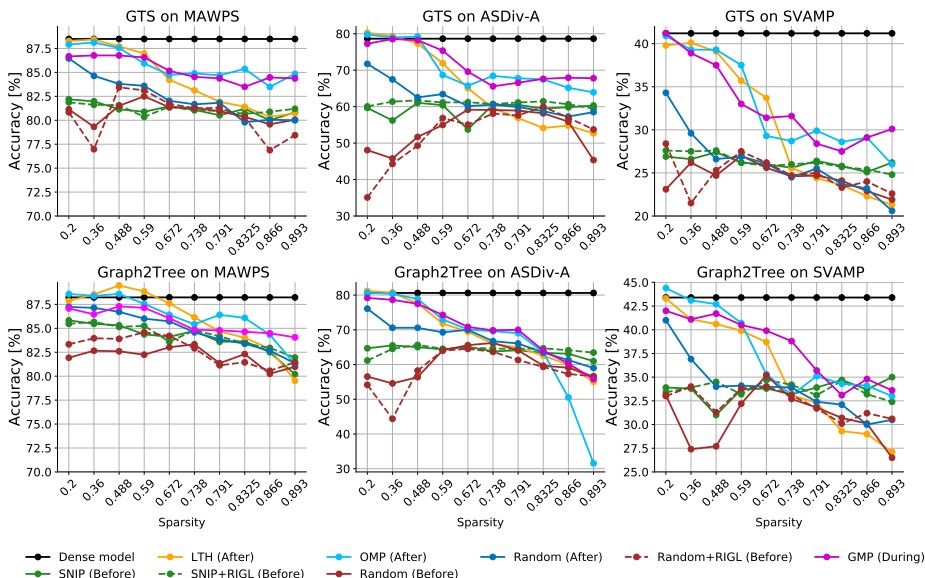

Figure 2: Arithmetic reasoning performance of various sparse GTS and Graph2Tree.

On MAWPS and ASDiv-A, models are trained with the training data and then evaluated on 5-fold cross-validation based on pre-assigned splits. For SVAMP, we train the models on a combination of MAWPS and ASDiv-A and test them on SVAMP, following Patel et al. (2021).

**Results and Analyses.** The performance on arithmetic reasoning is reported in Figure 2. The overall performance trend is very similar to the commonsense reasoning: SNNs can only match the dense performance when the sparsity level is lower than 48.8% with the exception of Graph2Tree on the relatively simple MAWPS dataset whose failing sparsity is 59%; SNNs are prone to sacrifice more performance on harder datasets (i.e., ASDiv-A and SVAMP) than the easier MAWPS dataset; prior-training methods perform no differently than random pruning. Moreover, we want to highlight that LTH surprisingly reaches lower accuracy than OMP and GMP at high sparsity levels, indicating that iterative magnitude pruning may not necessarily generalize better than on more complex tasks. Moreover, Magnitude Before Training (OMG (Before)) consistently causes severe layer collapse in the non-embedding layers, leading to zero accuracies. Since including the results of OMG (Before) will significantly dilute the performance difference of different sparsification methods, we choose to report it in Appendix B.

### 4.3 PROTEIN THERMAL STABILITY PREDICTION

#### 4.3.1 SEQUENCE-BASED MODELS

**Implementation Details.** We examine two classic sequence-based approaches in protein property prediction, *i.e.*, TAPE (Rao et al., 2019) and ESM-1B (Rives et al., 2021). For TAPE, we fine-tune it from the official pre-training (Rao et al., 2019) for 4 epochs with an initial learning rate of $1 \times 10^{-4}$ and a linear decay scheduler together with 100 warm-up steps. As for ESM-1B (Rives et al., 2021), starting from the official pre-trained checkpoint, we fine-tune the backbone with a learning rate of $1 \times 10^{-6}$ and the linear classification head on top of the backbone with a learning rate of $2 \times 10^{-2}$. The learning rate schedulers used for both backbone and linear head are OneCycle (Smith & Topin, 2019) decay schedulers. The training batch size is 2 for Meltome Atlas and 3 for HotProtein (HP-S). Classification accuracy on test sets is collected to measure the model performance.

**Results and Analyses.** In this section, we examine diverse sparse neural networks of sequence-based models (*i.e.*, transformers) on protein thermostability prediction benchmarks. ESM-1B (Rives et al., 2021), a SOTA approach in protein property modeling, is evaluated on both HotProtein (HP-S) and Meltome Atlas datasets. TAPE (Rao et al., 2019) is a classic transformer-based model adopted on HotProtein (HP-S). Extensive results of both static and dynamic sparsifications are collected in Figure 3. We observe that: ❶ For ESM-1B, all extracted sparse neural networks incur significant performance degradation whenever the sparsity level is larger than 20%. Note that here we only sparsify the fully connected layers in multi-head self-attentions & feed-forward networks of each

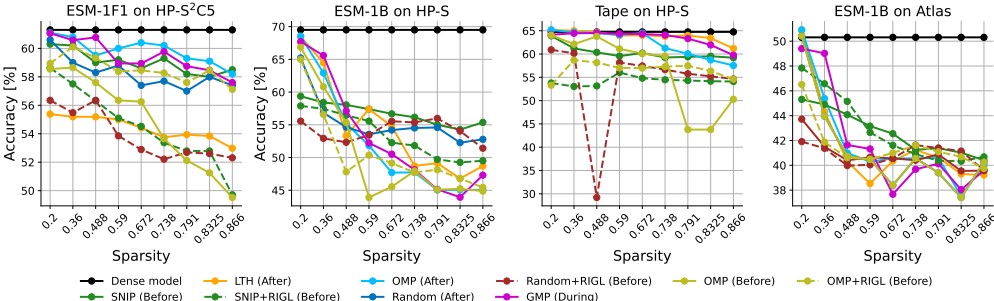

Figure 3: Protein prediction performance of various sparse models.

transformer layer, and leave all other modules unpruned. Even under this loose condition, ESM-1B still fails after pruning on both HP-S and Meltome Atlas, which indicates the low parameter redundancy in ESM-1B for modeling protein thermal stability. ❷ In general, static and dynamic pruning algorithms achieve similar performance with ESM-1B. SNIP (Before) and SNIP + RIGL (Before) deliver relatively better accuracies, especially for the high sparsity ($\geq 48.80\%$) subnetworks on HP-S. ❸ As for the worse backbone TAPE compared with ESM-1B, magnitude-based prunings like LTH (After), OMP (After), and GMP (During) show satisfactory results before $59\%$ sparsity.

Furthermore, we conduct a more fine-grained pruning schedule to investigate the tolerance of ESM-1B against sparsification. In detail, we prune $5\%$ weights with OMP (after) on different modules in ESM-1B and present the obtained accuracies in Table 1. Q, K, V, O, and FFN represent the fully connected layer in the query, key, value, & output of the self-attention module and feed-forward networks, respectively. The results show that whatever modules we select, $5\%$ **sparsity** damages the ESM-1B performance of protein thermostability prediction on HP-S$^2$C2.

Table 1: OMP (after) pruning $5\%$ weights of ESM-1B on different modules with HP-S$^2$C2.

| Pruned Modules | Accuracy ($\uparrow$) |
|---|---|
| None (Dense) | 94.68 |
| Q, K, V, O, FFN | 92.19 |
| Q, K, V | 92.55 |
| Q, K | 93.62 |
| Q, V | 93.62 |
| K, V | 92.55 |

### 4.3.2 STRUCTURE-BASED MODELS

**Implementation Details.** We further consider a representative structure-based algorithm for thermostability prediction, *i.e.*, ESM-IF1 (Hsu et al., 2022). Specifically, for ESM-IF1, we train the backbone and its linear head with learning rates of $1 \times 10^{-4}$ and $2 \times 10^{-2}$, respectively. A batch size of 4 is used for both models on HotProtein (HP-S$^2$C5). Classification testing accuracy is reported to reflect the model performance.

**Results and Analyses.** In this section, we study structure-based models and their sparse variants on HotProtein (HP-S$^2$C5). ESM-IF1 (Hsu et al., 2022), a recent SOTA approach, is chosen for benchmarking. It takes the 3D structure of proteins as input and predicts its thermal stable temperature. As shown in Figure 3, ESM-IF1 produces inferior sparse neural networks with all pruning mechanisms of both static and dynamic, where OMP (after) and GMP (During) present relatively better accuracies.

### 4.4 MULTILINGUAL TRANSLATION

**Implementation Details.** We choose the official multilingual model mBART[1] (Liu et al., 2020), which was originally pre-trained on 25 languages using masked language modeling (MLM), following the fine-tuning setting of Tang et al. (2020). We first choose 10 languages from the language pools used for MLM pre-training; create three sub-groups containing 2, 5, 10 languages; and fine-tune mBART on each sub-group, referring to 2-to-2, 5-to-5, and 10-to-10 multilingual translation fine-tuning, respectively. During inference, we report the averaged BLEU (Tang et al., 2020; Liu et al., 2020) scores of bi-directional translation across 10 languages to measure the translation performance. Hence, the task difficulty monotonically decreases from 2-to-2 to 5-to-5, and to 10-to-10 fine-tuning as more languages are involved during training. The default fine-tuning configurations in Tang et al. (2020) are adopted for 40K iterations with an Adam optimizer and a learning rate of $1 \times 10^{-6}$.

**Results and Analyses.** Intuitively, fewer languages involved during fine-tuning leads to a more difficult translation for all languages. As demonstrated in Figure 4, several consistent conclusions can be drawn: ❶ Besides OMP (After) and LTH (After), all other produced sparse subnetworks perform

---

[1]https://github.com/facebookresearch/fairseq

worse than the dense baseline when the sparsity is larger than 20%. The BLEU scores of OMP (After) and LTH (After) also decline and fail to match at $\geq 20\%$, $\geq 48.8\%$, $\geq 59\%$ sparsity levels for 2-to-2, 5-to-5, and 10-to-10 fine-tuning, respectively. ❷ Magnitude-based sparsifications like OMP, LTH, and GMP are comparably robust across all three translation settings, while other pruning methods have negligible advantages compared to random pruning. ❸ While the overall tendency of SNNs is quite consistent across different tasks, the prunability of mBART increases as more languages are involved during fine-tuning. It seems that multilingual translation has already been a challenging task for pruning, and involving more languages in inference causes extra obstacles. This is why in the hardest scenario of fine-tuning on 2-to-2 and evaluating with 10 languages, all sparse subnetworks suffer from substantial performance degradation.

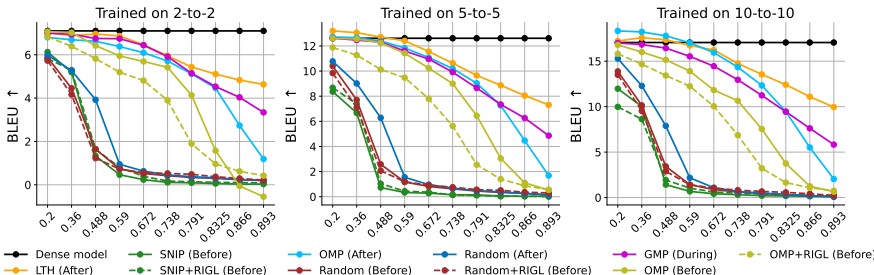

Figure 4: Multilingual performance of various sparse mBART. All models are tested on 10-to-10 multilingual translation and the averaged BLEU are reported.

## 4.5 Why SNNs Fail on SMC-Bench

We conduct a thorough investigation into the reasons why most SNNs struggle to perform on SMC-Bench. Our analysis identifies two possible causes for the failure: (1) the "lazy regime" in LLMs, and (2) the specific model components that are pruned. Based on these findings, we discover a set of stronger pruning recipes that alleviates (though still more or less suffering from) the failure on SMC-Bench, by breaking down the performance of the state-of-the-art BERT-pruning framework - oBERT (Kurtic et al., 2022) on SMC-Bench (note that most models evaluated in this paper are also BERT-based). Due to the limited space, we present our full investigation in Appendix C, and briefly present our sanity check of layer collapse below.

**Does layer collapse occur unexpectedly on SMC-Bench?** Layer collapse is the most common cause that blocks the information flow (signal propagation) of sparse neural networks, leading to a catastrophic performance drop (Tanaka et al., 2020). We plot the layerwise sparsity ratios of various sparse models in Appendix C.1. We do not observe severe layer collapse across methods except for SNIP which specifically removes nearly entire embedding layers. However, we do observe an unexpected phenomenon: *layerwise sparsities of different magnitude-based pruning approaches (i.e., IMP, OMP, and GMP) are extremely similar,* all overlapped on one line, despite the significant performance gap among them (up to 42.3%); small differences only start to appear until reaching very deep layers (e.g., classification heads) (see Appendix C.1 for more details). This abnormal behavior is highly correlated with the "lazy regime" (Neyshabur et al., 2020; Malladi et al., 2022) where the model stays in the same basin during fine-tuning with fairly small weight changes, and hence all magnitude-based pruning approaches, before, during, and after fine-tuning, tend to collapse to the same solution.

## 5 Conclusion

Given the enormous breakthroughs and the fruitful results that sparse neural networks have achieved in numerous fields, it is necessary to rethink the sufficiency of current evaluation protocols and introduce more difficult and diverse benchmarks to explore the limitation of sparse neural networks. In this paper, we assemble a large-scale, challenging, and more diverse benchmark, SMC-Bench. Through our thorough evaluation across various leading sparsifications, we confirm that SMC-Bench notably challenges the capability of most magnitude- or/and gradient-based sparse algorithms. We further dig deeper into the behavior of SNNs, and observe several surprising phenomena that are absent in the current evaluation. Our analysis points out that such failure is intimately related to the "lazy regime", which leads us to a suite of strong pruning recipes that alleviates (yet still more or less suffering from) the failure on SMC-Bench. Our subsequent effort will focus on exploring stronger sparse training algorithms that can scale and generalize on SMC-Bench, and meanwhile will consider the training costs of different sparse algorithms for a more holistic picture of their efficiency benefits.

## ACKNOWLEDGEMENT

We thank Dan Alistarh and Eldar Kurtic for the extremely helpful discussions about the implementation details of oBERT as well as our benchmark's claims; and Zhangheng Li for helping run extra experiments with oBERT. S. Liu and Z. Wang are in part supported by the NSF AI Institute for Foundations of Machine Learning (IFML). Part of this work used the Dutch national e-infrastructure with the support of the SURF Cooperative using grant no. NWO2021.060, EINF-2694 and EINF-2943/L1.

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

## A  SUMMARY OF TASKS, MODELS, DATASETS, AND TRAINING

We summarize the combinations of models and configurations that we used to evaluate SNNs on SMC-Bench.

Table 2: Summary of models and datasets that we used to evaluate on SMC-Bench.

| Task | Datasets | Models | Source |
|---|---|---|---|
| **Commonsense Reasoning** | CSQA | RoBERTa Large | Facebook AI Research Sequence-to-Sequence Toolkit (Ott et al., 2019) |
|  | WinoGrande | RoBERTa Large | Facebook AI Research Sequence-to-Sequence Toolkit (Ott et al., 2019) |
|  | RACE | RoBERTa Large | Facebook AI Research Sequence-to-Sequence Toolkit (Ott et al., 2019) |
| **Arithmetic Reasoning** | MAVPS | GTS | GitHub Repository (Patel et al., 2021) |
|  |  | Graph2Tree | GitHub Repository (Patel et al., 2021) |
|  | ASDiv-A | GTS | GitHub Repository (Patel et al., 2021) |
|  |  | Graph2Tree | GitHub Repository (Patel et al., 2021) |
|  | SVAMP | GTS | GitHub Repository (Patel et al., 2021) |
|  |  | Graph2Tree | GitHub Repository (Patel et al., 2021) |
| **Protein Thermostability Prediction** | HotProtein (HP-S) | TAPE | GitHub Repository (Rao et al., 2019) |
|  |  | ESM-1B | GitHub Repository (Rives et al., 2021) |
|  | HotProtein (HP-S$^2$C5) | ESM-IF1 | GitHub Repository (Rives et al., 2021) |
|  | HotProtein (HP-S$^2$C2) | ESM-1B | GitHub Repository (Rives et al., 2021) |
|  | Meltome Atlas | ESM-1B | GitHub Repository (Rives et al., 2021) |
| **Multilingual Translation** | 2-to-2 | mBART | Facebook AI Research Sequence-to-Sequence Toolkit (Ott et al., 2019) |
|  | 5-to-5 | mBART | Facebook AI Research Sequence-to-Sequence Toolkit (Ott et al., 2019) |
|  | 10-to-10 | mBART | Facebook AI Research Sequence-to-Sequence Toolkit (Ott et al., 2019) |

We strictly follow the training configurations reported in the original source to replicate the results of each task. The hyperparameters and configurations used for each model in this paper are shared below.

### A.1  COMMONSENSE REASONING

Table 3: Hyperparameters and training configurations used for models on Commonsense Reasoning.

| Models | **RoBERTa** | **RoBERTa** | **RoBERTa** |
|---|---|---|---|
| Dataset | CSQA | WinoGrande | RACE |
| Pre-trained Models | RoBERTa | RoBERTa | RoBERTa |
| Hidden Size | [1024] | [1024] | [1024] |
| FFN Inner Hidden Size | [4096] | [4096] | [4096] |
| Number of Layers | [24] | [24] | [24] |
| Learning Rate | [1e-5] | [1e-5] | [1e-5] |
| Weight Decay | [0.01] | [0.01] | [0.01] |
| Batch Size | [16] | [32] | [16] |
| Dropout | [0.1] | [0.1] | [0.1] |
| Attention Dropout | [0.1] | [0.1] | [0.1] |
| Clip Norm | [0.0] | [0.0] | [0.0] |
| Adam $\epsilon$ | [1e-06] | [1e-06] | [1e-06] |
| Adam $\beta_1$ | [0.9] | [0.9] | [0.9] |
| Adam $\beta_1$ | [0.98] | [0.98] | [0.98] |
| # Parameters | 355M | 355M | 355M |
| Training Time | 3000 steps | 23750 steps | 3 epochs |
| Wramup Time | 150 steps | 2375 steps | 500 steps |

## A.2 ARITHMETIC REASONING

Table 4: Hyperparameters and training configurations used for models on Arithmetic Reasoning.

| Models | GTS | Graph2Tree |
|---|---|---|
| Dataset | MAVPS, ASDiv-A, SVAMP | MAVPS, ASDiv-A, SVAMP |
| Pre-trained Embedding | RoBERTa | RoBERTa |
| Embedding Size | [768] | [768] |
| Hidden Size | [512] | [384] |
| Number of Layers | [2] | [2] |
| Learning Rate | [1e-3] | [8e-4] |
| Weight Decay | [1e-5] | [1e-5] |
| Embedding LR | [8e-6] | [1e-5] |
| Batch Size | [4 (MAVPS, ASDiv-A), 8 (SVAMP)] | [4 (MAVPS, ASDiv-A), 8 (SVAMP)] |
| Dropout | [0.5] | [0.5] |
| Adam $\epsilon$ | [1e-08] | [1e-08] |
| Adam $\beta_1$ | [0.9] | [0.9] |
| Adam $\beta_1$ | [0.999] | [0.999] |
| # Parameters | 140M | 143M |
| Training Time | 50 epochs | 50 epochs |

## A.3 PROTEIN THERMOSTABILITY PREDICTION

Table 5: Hyperparameters and training configurations used for models on Protein Thermostability Prediction.

| Models | TAPE | ESM-1B | ESM-IF1 |
|---|---|---|---|
| Dataset | HP-S | HP-S$^2$C2, Meltome Atlas, HP-S | HP-S$^2$C5 |
| Hidden Size | [768] | [1280] | [512] |
| Number of Layers | [12] | [33] | [20] |
| Learning Rate | [1e-4] | [2e-2 (head), 1e-6 (backbone)] | [2e-2 (head), 1e-4 (backbone)] |
| Weight Decay | [1e-2] | [1e-2] | [5e-2] |
| Batch Size | [16] | [3,2,3] | [4] |
| Attention Dropout | [0.1] | [0.0] | [0.1] |
| Dropout | [0.1] | [0.0] | [0.1] |
| Adam $\epsilon$ | [1e-08] | [1e-08] | [1e-08] |
| Adam $\beta_1$ | [0.9] | [0.9] | [0.9] |
| Adam $\beta_1$ | [0.999] | [0.999] | [0.999] |
| # Parameters | 92M | 650M | 124M |
| Training Time | 4 epochs | 4 epochs | 8 epochs |

## A.4 MULTILINGUAL TRANSLATION

Table 6: Hyperparameters and training configurations used for models on Multilingual Translation.

| Models | mBART | mBART | mBART |
|---|---|---|---|
| Dataset | 2-to-2 | 5-to-5 | 10-to-10 |
| Pre-trained Models | mBART | mBART | mBART |
| Hidden Size | [1024] | [1024] | [1024] |
| Number of Layers | [24] | [24] | [24] |
| Learning Rate | [3e-5] | [3e-5] | [3e-5] |
| Weight Decay | [0.0] | [0.0] | [0.0] |
| Batch Size | [16] | [32] | [16] |
| Dropout | [0.3] | [0.3] | [0.3] |
| Attention Dropout | [0.1] | [0.1] | [0.1] |
| Clip Norm | [0.0] | [0.0] | [0.0] |
| Adam $\epsilon$ | [1e-06] | [1e-06] | [1e-06] |
| Adam $\beta_1$ | [0.9] | [0.9] | [0.9] |
| Adam $\beta_1$ | [0.98] | [0.98] | [0.98] |
| # Parameters | 680M | 680M | 680M |
| Training Time | 40,000 steps | 40,000 steps | 40,000 steps |
| Wramup Time | 2,500 steps | 2,500 steps | 2,500 steps |

# B  RESULTS OF ARITHMETIC REASONING WITH MAGNITUDE BEFORE TRAINING

In this appendix, we report the performance of SNNs on arithmetic reasoning including Magnitude before Training (OMP (Before)). We can clearly observe that the accuracy of magnitude pruning before training dramatically falls from 80% to nearly 0% when sparsity is larger than 36%. After checking the corresponding layerwise sparsity, we find that OMP (Before) completely removes all the weights from non-embedding and non-encoder layers, leading to severe layer collapse.

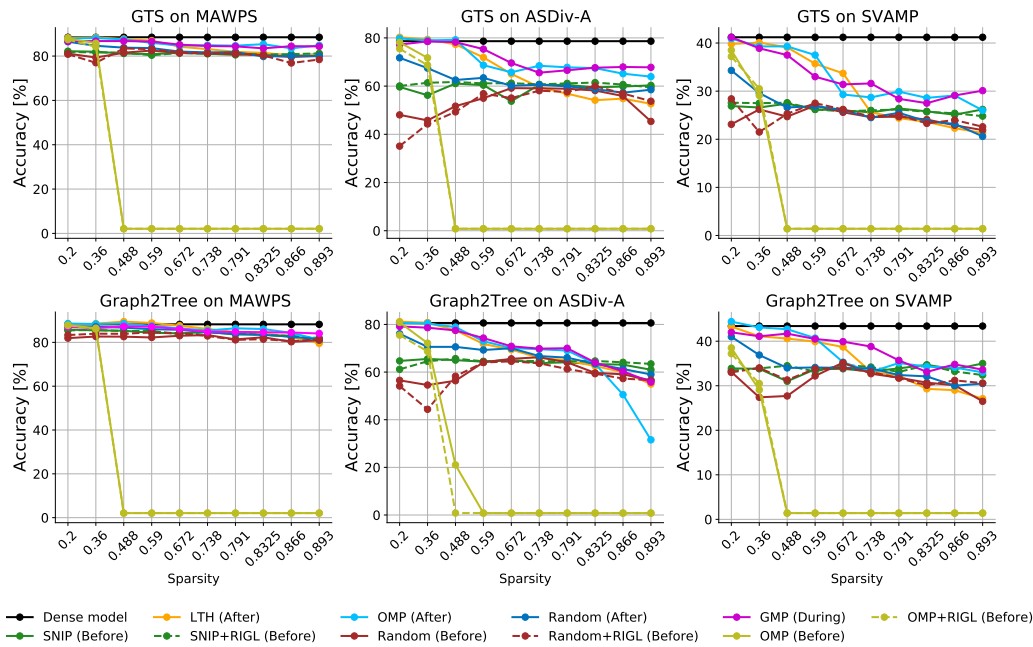

Figure 5: Arithmetic reasoning performance of various sparse GTS and Graph2Tree on MAWPS, ASDiv-A, and SVAMP.

## C  AN INVESTIGATION OF WHY SNNS FAIL ON SMC-BENCH

In this section, we conduct a full investigation, attempting to open the box for the potential causes of SNN failures on SMC-Bench. Our analysis reveals two possible causes: (1) the "lazy regime" in fine-tuning LLMs, and (2) the model components to prune. Due to the "lazy regime" phenomenon (Chizat et al., 2019; Malladi et al., 2022), commonly used pruning techniques that rely on magnitude and gradient can be very uninformative. Therefore, we turn to the latest strong second-order pruning framework - oBERT (Kurtic et al., 2022), which utilizes inverse-Hessian approximations to guide pruning decisions. We choose RoBERTa.large on CSQA to conduct this investigation. The roadmap of our full investigation is presented below.

- **Layerwise sparsity ratios** of various magnitude-based pruning methods are strikingly similar, suggesting that "lazy regime" may occur during fine-tuning. In this regime, the most common pruning criteria such as magnitude and gradient can be rather unreliable.

- **Second-order pruning** approaches like oBERT provide more faithful signals than magnitudes and gradients for LLM pruning, and hence achieve significantly higher accuracy at high sparsities.

### C.1  LAYERWISE SPARSITY RATIOS ON SMC-BENCH

To check if severe layer collapse occurs on SMC-Bench, we plot the per-layer sparsity ratios discovered by various sparsification approaches at three sparsity levels: 36%, 64%, and 83%. Layers are ordered from input to output on the X-axis. We respectively report the layerwise sparsity of commonsense reasoning with RoBERTa on CSQA and RACE in Figure 6 and 7, and arithmetic reasoning with GTS on SVAMP in Figure 8. We summarize our main findings here.

❶ **Layerwise sparsities of magnitude-based pruning approaches are extremely similar.** IMP, OMP, and GMP that rely on weight magnitude for pruning share an extremely similar set of layerwise sparsities. Especially, sparsity values of magnitude pruning on commonsense reasoning are completely identical, all overlapped on blue lines, except for the tiny difference in classification heads. This phenomenon indicates that weights of RoBERTa excluding classifiers remain rather stable during commonsense reasoning fine-tuning so that all the magnitude pruning variants (both before and after) discover the same sparsity pattern. The sparsity difference of arithmetic reasoning is more distinguishable than commonsense reasoning. Still, sparsities in the encoder (pre-trained RoBERTa) of IMP, GMP, and OMP (After) largely overlap. Until reaching the later layers, the sparsity ratios of different approaches start to be distinct. ❷ **SNIP tends to prune all the weights in embedding layers aggressively.** Even at the mild 36% sparsity, SNIP prunes weights of embedding layers to 99.7% sparsity, which may explain why SNIP struggles on SMC-Bench. ❸ **OMP (Before) suffers from layer collapse on arithmetic reasoning.** We empirically find that OMP (Before) leads to completely empty deep layers when sparsity is larger than 36%, indicating the limitation of only considering pruning with the magnitude before fine-tuning or re-training.

The near-identical layerwise sparsity ratios across various magnitude-based methods remind us of the "lazy training" regime (Chizat et al., 2019; Malladi et al., 2022) which was revealed to occur during the fine-tuning of LLMs. Under this regime, weight changes during fine-tuning are negligible, hence non-informative and more "noisy". Consequently, various magnitude-based pruning approaches, regardless of their timing, all tend to converge to the same solution.

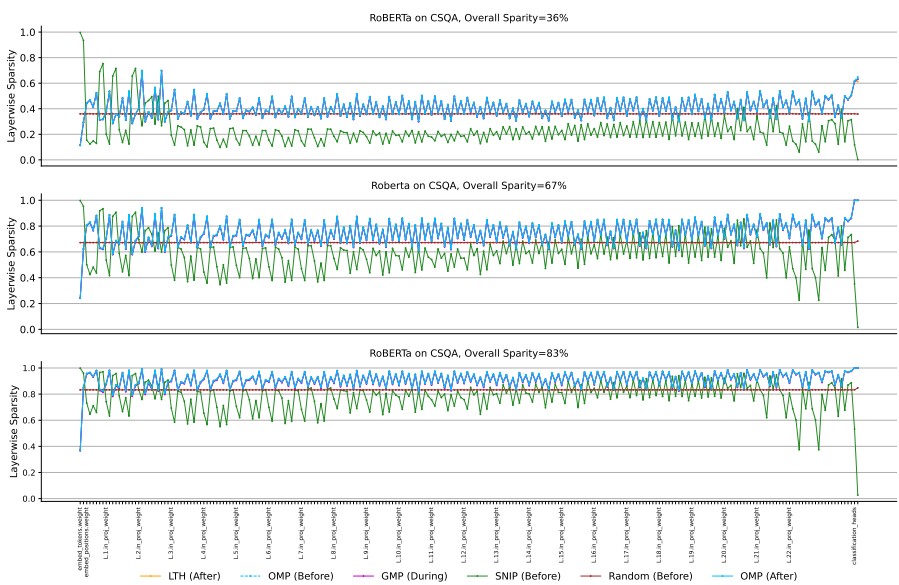

Figure 6: Layerwise sparsity of RoBERTa on CSQA at sparsity levels $\in [36\%, 67\%, 83\%]$.

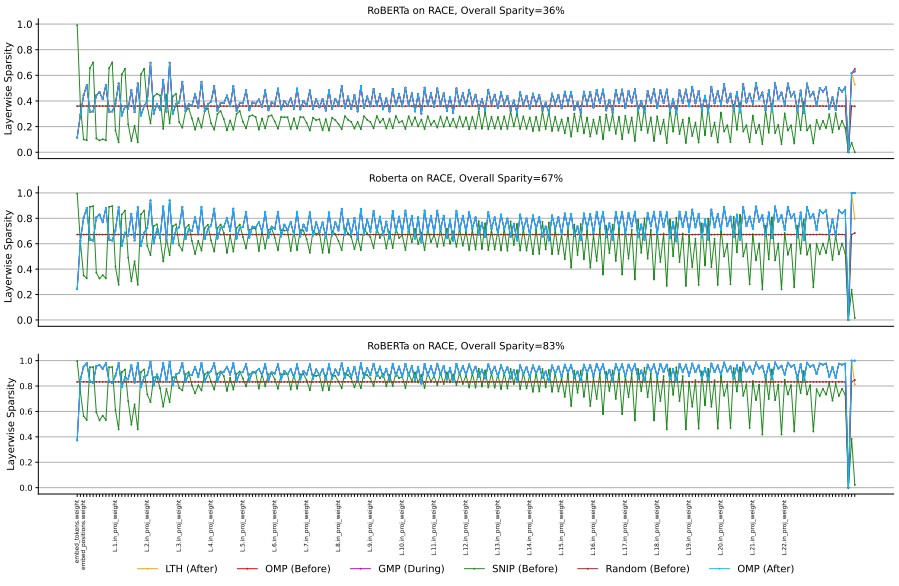

Figure 7: Layerwise sparsity of RoBERTa on RACE at sparsity levels $\in [36\%, 67\%, 83\%]$.

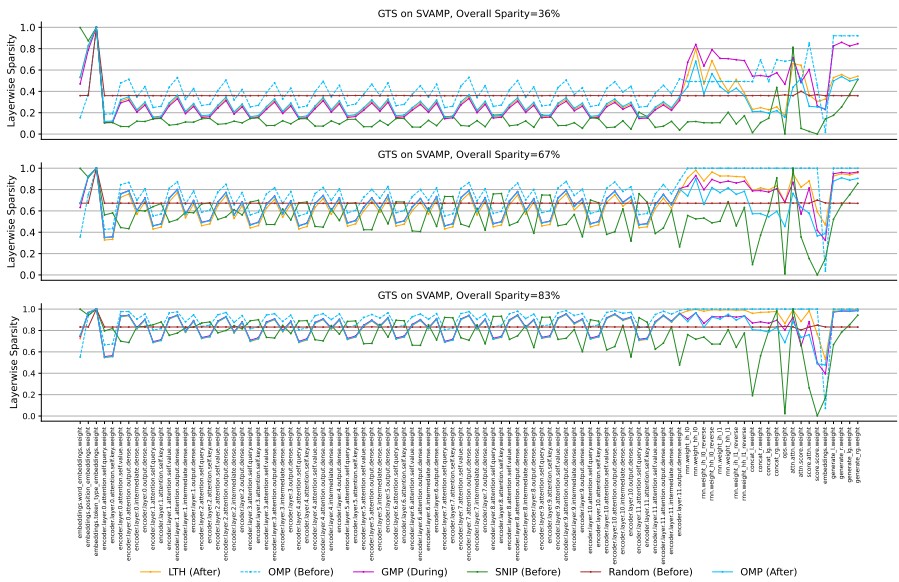

Figure 8: Layerwise sparsity of GTS on SVAMP at sparsity levels $\in [36\%, 67\%, 83\%]$.

## C.2 EVALUATION OF OBERT ON SMC-BENCH

Based on our conjecture that magtitudes/gradients become unreliable for pruning LLMs, we hypothesize that the second-order pruning approaches with approximated Hessian matrix would be more accurate options. To verify our conjecture, we turn to the latest stronger second-order pruning framework, oBERT (Kurtic et al., 2022). Specifically, we follow Kurtic et al. (2022) and replace the magnitude pruning criterion of the LTH framework with the second-order oBERT criterion; adopt Learning Rate Rewinding (LRR) (Renda et al., 2020) and Knowledge Distillation (KD) (Hinton et al., 2015) during each pruning iteration; and keep the embeddings and classification heads dense.

Figure 9 support our hypothesis and demonstrates that oBERT notably outperforms the zero-order and first-order sparsification approaches, and substantially improves the accuracy to a competitive level. More importantly, oBERT produces a completely different layerwise sparsity pattern from magnitude-based pruning approaches, which is consistent with the patterns that are commonly observed in sparse computer vision models: deeper layers tend to have higher sparsities than lower layers (Evci et al., 2020; Kusupati et al., 2020; Tanaka et al., 2020; Liu et al., 2021b).

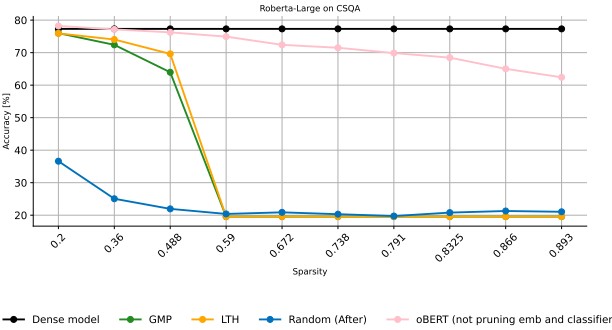

Figure 9: A roadmap of accuracy recovering via a suite of stronger pruning recipes with RoBERTa-Large on CSQA.

So far, our investigation has discovered that the pruning recipe used in (Kurtic et al., 2022) can remarkably improve the LLM pruning performance on SMC-Bench. Nevertheless, such a well-tuned pruning recipe is both time- and resource-intensive ($9\times$ more fine-tuning time, besides Hessian matrix

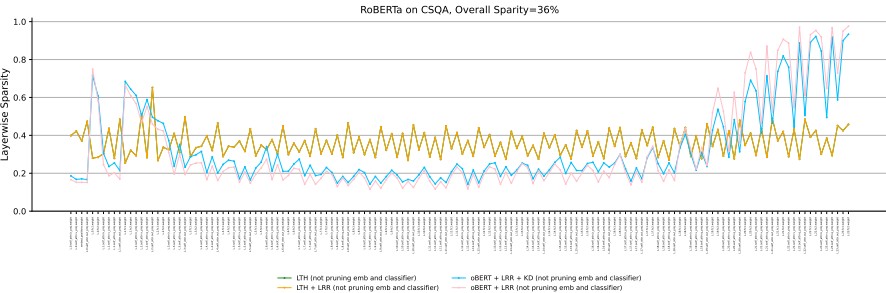

Figure 10: Layerwise sparsity comparison among LTH, LRR, oBERT, and KD with RoBERTa-Large on CSQA.

approximation); and even so, the strongest SNNs still fall short of their dense counterpart by around 10% accuracy at sparsities between $60\% - 80\%$, in contrast to "normal" SNNs that easily match their dense models on CIFAR, ImageNet, or GLUE. Therefore, the main claim of our paper still holds, that is, SMC-Bench indeed provides a new benchmark that is way more challenging for SOTA sparse algorithms, than existing testbeds.

# D SUMMARY OF EVALUATION TASKS AND DATASETS IN 100 PAPERS

Table 7: Summary of Evaluation Tasks and Datasets Used in 100 Recent SNN Papers.

| TASK | TOTAL #PAPER | DATASETS | #PAPER |
|---|---|---|---|
| IMAGE CLASSIFICATION | 82 | IMAGENET | 62 |
| | | CIFAR-10 | 59 |
| | | CIFAR-100 | 37 |
| | | MNIST | 26 |
| | | FASHION MNIST | 10 |
| | | SVHN | 4 |
| | | BIRDS-200 | 1 |
| | | FLOWERS-102 | 1 |
| | | EMNIST | 1 |
| NLP TASK | 16 | GLUE | 9 |
| | | SQUAD | 4 |
| | | WIKITEXT-103 | 3 |
| | | WMT | 5 |
| | | IMDB | 1 |
| | | AAN | 1 |
| | | LO | 1 |
| | | OPENWEB TEXT | 1 |
| | | ONE BILLION WORD BENCHMARK | 1 |
| FACE RECOGNITION | 3 | LFW | 3 |
| | | YOUTUBE FACES | 2 |
| | | CASIA-WEBFACE | 1 |
| OBJECT DETECTION | 3 | COCO DATASET | 2 |
| | | PASCAL-VOL-2007 | 1 |
| SPEECH RECOGNITION | 2 | GOOGLE-12 | 1 |
| | | TIMIT | 1 |
| HIGH-RESOLUTION RECONSTRUCTION | 2 | SET5 | 2 |
| | | SET14 | 2 |
| | | B100 | 2 |
| | | URBAN100 | 2 |
| | | MANGA109 | 2 |
| IMAGE GENERATION | 2 | CIFAR-10 | 2 |
| | | IMAGENET | 1 |
| | | STL-10 | 1 |
| HUMAN ACTIVITY RECOGNITION | 1 | HAR-2 | 1 |
| MICROARRAY CLASSIFICATION | 1 | LEUKEMIA | 1 |
| | | CLL-SUB-111 | 1 |
| | | SMK-CAN-18 | 1 |
| | | GLI-85 | 1 |
| HAND GESTURE RECONSTRUCTION | 1 | NVGESTURE | 1 |
| REGRESSION TASK | 1 | NYU DEPTH | 1 |
| 3D OBJECT PART SEGMENTATION | 1 | SHAPENET | 1 |
| RL TASK | 1 | CARTPOLE | 1 |
| | | ACROBOT | 1 |
| | | MOUNTAINCAR | 1 |
| | | ATARI SUITE | 1 |
| VEDIO DEBLURRING | 1 | DVD | 1 |
| | | GOPRO | 1 |
| | | REAL BLURRY VIDEOS | 1 |
| VOCABULARY SPEECH RECOGNITION | 1 | VS | 1 |
| | | SWB | 1 |

