# OpenReview forum: "Sparsity May Cry: Let Us Fail (Current) Sparse Neural Networks Together!"
_ICLR.cc/2023/Conference — ICLR 2023 notable top 25%_

### Official Review · Reviewer_yFvQ · 2022-10-25

**Confidence:** 4
**Correctness:** 3
**Technical Novelty And Significance:** 3
**Empirical Novelty And Significance:** 4
**Recommendation:** 8

**Clarity, Quality, Novelty And Reproducibility:**

Writing is very clear. High technical quality and novelty as a “new perspective” benchmark in sparsity and pruning.

**Strength And Weaknesses:**

Overall, I enjoy this work a lot and think it will make a big/good splash in the sparsity community. But I also believe its scientific rigor needs to be enhanced in several aspects before being treated as an indeed solid benchmark. See details below.

First, it might be too early to declare sparsity will cry! For example, only first-order, unstructured pruning has been so far tested, which represent a strong yet still restricted subclass of modern pruning methods. I would definitely suggest the authors to pick representatives from other pruning families, such as second-order pruning.

Second, how the authors could justify their chosen datasets and tasks are indeed more challenging than ImageNet? IMHO, the sole fact that current pruning methods work better on ImageNet than those cannot directly support the claim, since it could also be some task types that are just more “pruning (un)friendly”.

Similarly, how the authors can justify their chosen models (admittedly very large already) are sufficiently “overparameterized” for those new difficult tasks? For example, would it be possible that, if continuing to increase the model size on those tasks, the generalization performance will keep improving, while pruning algorithms also start to gain back their effectiveness?


**Summary Of The Paper:**

This paper introduces “Sparsity May Cry” Benchmark (SMC-Bench), a collection of carefully curated 4 diverse tasks with 12 datasets, for a more general evaluation and unveiling the true potential of sparse algorithms. Evaluation reveals several important and unusual findings.

**Summary Of The Review:**

Overall, I would tend to accept this paper. If the author can address my concerns. I would be more convinced.

---

> ### Author Response · Authors · 2022-11-15
> **Response to Reviewer yFvQ (2/2):**
>
> **Comment 2: Second, how the authors could justify their chosen datasets and tasks are indeed more challenging than ImageNet? IMHO, the sole fact that current pruning methods work better on ImageNet than those cannot directly support the claim, since it could also be some task types that are just more “pruning (un)friendly”.**
>
> * Good point! We think it is rather difficult to directly prove that the tasks in SMC-Bench are more challenging than the ImageNet given they are from different fields. Here, we choose to evaluate their difficulties based on the performance gaps between their ceiling human performance and the best performance achieved by machine learning approaches, which we believe is a relatively fair way to justify their difficulties.
>
> * Commonsense reasoning and arithmetic reasoning have been considered as challenging tasks in the field of NLP for a long time, where scaling up model size alone has not proved sufficient for achieving high performance [2,3,4], even with the advanced prompting technique. For instance, SOTA NLP models with hundreds of billions of parameters are still far away from the ceiling human performance as shown below (the higher the better) [2].  Overall,  large language models are less capable of all of these things than human beings with modest domain knowledge.
> | Models (#Param.)  | Gopher (280B)  | GPT-3 (175B) | Megatron-Turing (530B) | Human Ceiling |
> | ------------- | :-----------: |:-----------:|:-----------: |:-----------: |
> | RACE-h |  71.6 |  46.8 | 47.9 | 94.2 |
>
> * On the other hand, the error rate of a human on CIFAR-10 is estimated to be around 6% which a naive ResNet-20 model can easily beat. The top-5 human accuracy of ImageNet, which is generally considered as a complex enough task in computer vision, is 5.1% [5], which has been already surpassed by deep learning algorithms early in 2015 [6]. We hope in this way we can say that the tasks collected by SMC-Bench are more challenging than ImageNet.
>
> **Comment 3: Similarly, how the authors can justify their chosen models (admittedly very large already) are sufficiently “overparameterized” for those new difficult tasks? For example, would it be possible that, if continuing to increase the model size on those tasks, the generalization performance will keep improving, while pruning algorithms also start to gain back their effectiveness?"**
>
> * We never claim that we choose the models that are overparameterized enough, instead, we choose the models that are commonly used in the field, as well as our GPUs can fit. You are correct that if we continue to increase the model size on those tasks the generalization performance will keep improving, as we have seen such trends many times recently [3,4,7]. However, it remains unclear if the performance of the pruning algorithms will also start to gain back their effectiveness, as even the dense models with hundreds of billions of parameters seem do not suffer from overfitting as model size scaling on these challenging tasks [3,4,7].
>
> [1] Kurtic, Eldar, et al. "The Optimal BERT Surgeon: Scalable and Accurate Second-Order Pruning for Large Language Models." arXiv preprint arXiv:2203.07259 (2022).
>
> [2] Rae, Jack W., et al. "Scaling language models: Methods, analysis and insights from training gopher." arXiv preprint arXiv:2112.11446 (2021).
>
> [3] Wei, Jason, et al. "Chain of thought prompting elicits reasoning in large language models." arXiv preprint arXiv:2201.11903 (2022).
>
> [4] Srivastava, Aarohi, et al. "Beyond the Imitation Game: Quantifying and extrapolating the capabilities of language models." arXiv preprint arXiv:2206.04615 (2022).
>
>
> [5] Russakovsky, Olga, et al. "Imagenet large scale visual recognition challenge." International journal of computer vision 115.3 (2015): 211-252.
>
> [6] He, Kaiming, et al. "Delving deep into rectifiers: Surpassing human-level performance on imagenet classification." Proceedings of the IEEE international conference on computer vision. 2015.
>
> [7] Chowdhery, Aakanksha, et al. "Palm: Scaling language modeling with pathways." arXiv preprint arXiv:2204.02311 (2022).

---

> > ### Comment · Reviewer_yFvQ · 2022-11-17
> > **Thanks for the detailed rebuttal**
> >
> > My concerns have been well addressed. Thus, I increase my score.

---

> > > ### Author Response · Authors · 2022-11-27
> > > **Thank you for your support!**
> > >
> > > We thank you again for all the helpful feedback and very positive evaluation. We really appreciate it for increasing your score. Your support means a lot to us!
> > >
> > > Best wishes,
> > >
> > > Authors

---

> ### Author Response · Authors · 2022-11-15
> **Response to Reviewer yFvQ (1/2):**
>
> We are very grateful for your positive score, your agreement with our contribution, and your graceful compliments.
>
> **Comment 1: First, it might be too early to declare sparsity will cry! For example, only first-order, unstructured pruning has been so far tested, which represent a strong yet still restricted subclass of modern pruning methods. I would definitely suggest the authors to pick representatives from other pruning families, such as second-order pruning.**
>
> * As you suggested, we evaluated the recently proposed SOTA second-order pruning method Optimal BeRT Surgeon (oBERT) [1] due to its promising expertise on BERT pruning (our models on Complex Reasoning tasks are also BERT based - RoBERTa). The results are reported in below Table. Due to the memory issue, we test it with RoBERTa-based instead of RoBERTa-large which is used in our submission. We evaluated on the Roberta base model instead of large due to the OOM issue. The hyperparameters were chosen as the follows: we adjust the num_grads as the total number of iterations in one epoch (1264 for Winogrande and 600 for CSQA), Fisher_block_size is set as B=50, and lamda is 10-7. We can see that while oBERT seems is a better pruning method than LTH and GMP, it also suffers from a dramatic performance loss. Therefore while the entire big picture remains open to explore, we tend to conjecture that our claim still holds in general for second-order pruning methods.
> * **RoBERTa-base on commonsense QA**:
> | Sparsity  | 0.2 | 0.36 | 0.48 | 0.59 | 0.67 | 0.73 | 0.79 | 0.83 | 0.86 | 0.89 |
> | ------------- | :-----------: |:-----------:|:-----------: |:-----------: |:-----------: |:-----------:|:-----------: |:-----------: |:-----------: |:-----------:|
> | Dense | 66.91 | 66.91 | 66.91 | 66.91 | 66.91 | 66.91 | 66.91 | 66.91 | 66.91 | 66.91 |
> | LTH  | **66.91**|  64.94|  59.37|  32.84| 19.57|  19.57|  19.57|  19.57|  **19.57**| 19.57|
> | GMP |64.78| 56.76| 43.49| 19.57| 19.57| 19.57| 19.57| 19.57| **19.57**| 19.57|
> |oBERT | 66.50| **65.11**| **62.00**| **52.91**| **37.10** | **25.88**| **22.19**| **19.90**| 16.54| **20.07**|
> * **RoBERTa-base on WinoGrande**:
> | Sparsity  | 0.2 | 0.36 | 0.48 | 0.59 | 0.67 | 0.73 | 0.79 | 0.83 | 0.86 | 0.89 |
> | ------------- | :-----------: |:-----------:|:-----------: |:-----------: |:-----------: |:-----------:|:-----------: |:-----------: |:-----------: |:-----------:|
> | Dense | 68.09 | 68.09| 68.09| 68.09| 68.09| 68.09| 68.09| 68.09| 68.09| 68.09|
> | LTH  | **67.17** | **65.72** | 61.99 | 56.55 |**54.75** | **51.93** | **51.77** | **51.65** |51.00 | 50.22|
> | GMP |67.02|65.27| **62.14**|56.78|52.82| 50.33| 50.61| 51.07| **51.98**|49.64|
> |oBERT | 66.64|**65.72**|61.38| **57.24**| 52.51| 51.06| 50.33|50.55| 50.46| **50.45**|

---

### Official Review · Reviewer_wzLQ · 2022-10-25

**Confidence:** 4
**Correctness:** 3
**Technical Novelty And Significance:** 4
**Empirical Novelty And Significance:** 4
**Recommendation:** 8

**Clarity, Quality, Novelty And Reproducibility:**

Clarity: easy to read and understand

Quality: solid and comprehensive work

Novelty: high

Reproducibility: good (the authors promised to open-source)


**Strength And Weaknesses:**

Strength:
- This paper raises a very interesting new angle on the uprising field of sparse neural networks. Given the enormous that sparse neural networks have achieved, the authors ask a timely question: are current benchmarks sufficiently challenging for them, and (if indeed so) are we too optimistic in declaring the prevailing win of sparsity? The authors choose a few latest datasets and tasks that require substantial reasoning and structure information. This new benchmark will make an exciting addition to several existing sparse NN benchmarks such as Frankle et al. (2021).

- Comparing a large range of before-training, during-training and after-training pruning methods (including both static and dynamic), the authors observe several surprising phenomena that are absent in the current evaluation. Most alarmingly, all of the SOTA sparse algorithms bluntly fail to perform on SMC-Bench, sometimes at significantly trivial sparsity e.g., 5%. The failure does not appear specific to one approach but unanimously across all sparse NN approaches that were evaluated. That points us to some rather fundamental rethinking of both sparse NN training and evaluation currently.

- The authors also identified that model “prunability” is intimately related to task difficulty: those models trained on difficult tasks suffer more from pruning compared to easier tasks. Also, some best pruning algorithms start to perform indistinguishably from the random pruning baseline on the new challenging benchmark, and iterative pruning does not necessarily generalize better than one-shot pruning either. All those findings offer brand-new knowledge of broad interest to the sparse NN community.

Weaknesses:
- The paper did not identify a root cause for the failure of SNNs on SMC-Bench, although exploring then excluding a few factors. I feel one main reason is probably intimately related to the “lazy training” dynamics emerging in sufficiently overparameterized models, e.g., after training weights will only change very little from initialization.
Moreover, in one of their sanity checks on whether pruning embedding layers or not, the authors found SNIP to significantly underperform others because it prunes too much of the pre-trained embedding while others not. I am wondering whether we could ad-hoc skip that embedding, for all pruning methods and then compare them?

- I strongly suggest the authors to add to their comparison one or more second-order pruning methods. such as Optimal Brain Damage, Fisher Pruning, or the recent Optimal BeRT Surgeon.

- Also missed in the current benchmark is the structured pruning family: will they be more robust or fragile when task difficulty and data volume scale up? How about sparse MoE methods?


**Summary Of The Paper:**

In this paper, the authors assemble a large-scale, challenging, and more diverse benchmark, SMC-Bench, for pruning and sparse training algorithms. A careful evaluation indicates that SMC-Bench significantly challenges current SOTA sparse algorithms, exposes their limitations and points to new research opportunities.

**Summary Of The Review:**

This paper proposes an interesting method and the major concerns lie on more comparisons in the experiments.

---

> ### Author Response · Authors · 2022-11-15
> **Response to Reviewer wzLQ (2/2)**
>
> **Comment 3: Also missed in the current benchmark is the structured pruning family: will they be more robust or fragile when task difficulty and data volume scale up? How about sparse MoE methods?**
>
> * Following [3,4], we have evaluated the commonly used structured pruning algorithm, $\ell_1$ norm structured pruning from Li et al. [5]. We observe that that structured pruning is significantly more fragile than unstructured pruning on SMC-Bench, sometimes completely fails. Perhaps, more advanced structured pruning approaches are required to perform on SMC-Bench.
> * **RoBERTa-base on commonsense QA**:
> | Sparsity  | 0.2 | 0.36 | 0.48 | 0.59 | 0.67 | 0.73 | 0.79 | 0.83 | 0.86 | 0.89 |
> | ------------- | :-----------: |:-----------:|:-----------: |:-----------: |:-----------: |:-----------:|:-----------: |:-----------: |:-----------: |:-----------:|
> | Dense | 68.09 | 68.09| 68.09| 68.09| 68.09| 68.09| 68.09| 68.09| 68.09| 68.09|
> | Structured L1 Norm | 19.5|19.5|19.5|19.5|19.5|19.5|19.5|19.5|19.5|19.5|
>
> * Although sparse MoE is an exciting direction for model scaling via sparsely-activated experts, they are beyond the scope of this work. Our goal was to study the growing literature on finding data-independent unstructured subnetworks with sparse weight matrix, and to highlight the limitations of the current evaluation protocols; sparse MoE is data-dependently and sparsely activates experts (such as MLPs or filters) in a structured level not the unstructured level, so it does not fall into this category. Yet, we are absolutely interested in evaluating sparse MoE on SMC-Bench in the future given its outstanding performance in model scaling.
>
> [1] Malladi, Sadhika, et al. "A Kernel-Based View of Language Model Fine-Tuning." arXiv preprint arXiv:2210.05643 (2022).
>
> [2] Kurtic, Eldar, et al. "The Optimal BERT Surgeon: Scalable and Accurate Second-Order Pruning for Large Language Models." arXiv preprint arXiv:2203.07259 (2022).
>
> [3] Liu, Shiwei, et al. "Sparse training via boosting pruning plasticity with neuroregeneration." NeurIPS 2021.
>
> [4] Renda, Alex, Jonathan Frankle, and Michael Carbin. "Comparing rewinding and fine-tuning in neural network pruning." ICLR 2020.
>
> [5] Li et al. Pruning Filters for Efficient ConvNets. ICLR, 2017.

---

> ### Author Response · Authors · 2022-11-15
> **Response to Reviewer wzLQ (1/2)**
>
> **Comment 1: The paper did not identify a root cause for the failure of SNNs on SMC-Bench, although exploring then excluding a few factors. I feel one main reason is probably intimately related to the “lazy training” dynamics emerging in sufficiently overparameterized models, e.g., after training weights will only change very little from initialization. Moreover, in one of their sanity checks on whether pruning embedding layers or not, the authors found SNIP to significantly underperform others because it prunes too much of the pre-trained embedding while others not. I am wondering whether we could ad-hoc skip that embedding, for all pruning methods and then compare them?**
>
> * Thank you for your suggestions! We agree with your conjecture that the "lazy training" regime might be a cause for the failure of SNNs on SMC-Bench. Theoretically, the "lazy training'' regime has been extended to the language model fine-tuning scenario [1] proving that only ``small changes'' are required to solve downstream tasks. The practical evidence can also be found in our paper Appendix C where we show that different magnitude-based methods surprisingly lead to very similar layer-wise sparsity patterns even though the pruning occurs at totally different times. This also implies that negligible changes in weights have happened during fine-tuning. However, our arithmetic reasoning tasks do not perform fine-tuning but are trained from scratch. The "lazy training'' theory solely may not be sufficient to fully explain our results.
>
>
> * Regarding your suggestions about the skip of embedding layers, we actually have done it as shown in Figure 5 in our original submission. This ad-hoc embedding skipping can slightly mitigate the failures of SNNs on SMC-Bench, but can not fully address them.
>
> **Comment 2: I strongly suggest the authors to add to their comparison one or more second-order pruning methods. such as Optimal Brain Damage, Fisher Pruning, or the recent Optimal BeRT Surgeon.**
>
> * As you asked, we evaluated the recently proposed SOTA second-order pruning method Optimal BeRT Surgeon (oBERT) [2] due to its promising expertise on BERT pruning (our models on Complex Reasoning tasks are also BERT based - RoBERTa). We evaluated on the Roberta base model instead of large due to the OOM issue. The hyperparameters were chosen as the follows: we adjust the num_grads as the total number of iterations in one epoch (1264 for Winogrande and 600 for CSQA), Fisher_block_size is set as B=50, and lamda is 10-7. The results are reported in below Table. Due to the memory issue, we test it with RoBERTa-based instead of RoBERTa-large which is used in our submission.  We can see that while oBERT seems is a better pruning method than LTH and GMP, it also suffers from a dramatic performance loss. Therefore, we believe our claim still holds for second-order pruning methods.
> * **RoBERTa-base on commonsense QA**:
> | Sparsity  | 0.2 | 0.36 | 0.48 | 0.59 | 0.67 | 0.73 | 0.79 | 0.83 | 0.86 | 0.89 |
> | ------------- | :-----------: |:-----------:|:-----------: |:-----------: |:-----------: |:-----------:|:-----------: |:-----------: |:-----------: |:-----------:|
> | Dense | 66.91 | 66.91 | 66.91 | 66.91 | 66.91 | 66.91 | 66.91 | 66.91 | 66.91 | 66.91 |
> | LTH  | **66.91**|  64.94|  59.37|  32.84| 19.57|  19.57|  19.57|  19.57|  **19.57**| 19.57|
> | GMP |64.78| 56.76| 43.49| 19.57| 19.57| 19.57| 19.57| 19.57| **19.57**| 19.57|
> |oBERT | 66.50| **65.11**| **62.00**| **52.91**| **37.10** | **25.88**| **22.19**| **19.90**| 16.54| **20.07**|
> * **RoBERTa-base on WinoGrande**:
> | Sparsity  | 0.2 | 0.36 | 0.48 | 0.59 | 0.67 | 0.73 | 0.79 | 0.83 | 0.86 | 0.89 |
> | ------------- | :-----------: |:-----------:|:-----------: |:-----------: |:-----------: |:-----------:|:-----------: |:-----------: |:-----------: |:-----------:|
> | Dense | 68.09 | 68.09| 68.09| 68.09| 68.09| 68.09| 68.09| 68.09| 68.09| 68.09|
> | LTH  | **67.17** | **65.72** | 61.99 | 56.55 |**54.75** | **51.93** | **51.77** | **51.65** |51.00 | 50.22|
> | GMP |67.02|65.27| **62.14**|56.78|52.82| 50.33| 50.61| 51.07| **51.98**|49.64|
> |oBERT | 66.64|**65.72**|61.38| **57.24**| 52.51| 51.06| 50.33|50.55| 50.46| **50.45**|

---

### Official Review · Reviewer_JUeu · 2022-10-26

**Confidence:** 4
**Correctness:** 4
**Technical Novelty And Significance:** 2
**Empirical Novelty And Significance:** 3
**Recommendation:** 8

**Clarity, Quality, Novelty And Reproducibility:**

The manuscript is mostly very clear - tho, as mentioned above, I'd expect a bit more details (or better organized/prezented) list of used benchmarks, and more info about open-sourcing.

The reproducibility is limited with the current state - but hopefully will be trivial after the authors open-source.

**Strength And Weaknesses:**

The paper attaches an obvious question that seems not to have been addressed yet.  It's obvious that the authors spent an enormous amount of time and effort to collect all the results.

The authors promise to open-source their dataset, however no more details are provided. Additionally one would expect a well organized list of the "10 open source repositories" used for creating the benchmark. The open-sourcing and hopefully stream-lining future benchmarks is the most valuable part of the authors' work.

I would also expect the authors to evaluate the networks on the "weak"/"easy" datasets - to confirm that their implementations/runs are consistent with what was claimed in the intro

**Summary Of The Paper:**

The paper introduces a new set of benchmarks dubbed "SMC-Bench" (Sparsity May Cry). The authors correctly remark that the ever-growing research in sparsifying neural networks is often (if not always) evaluated on limited set of benchmarks which are often one of the very well understood, and "relatively easy" datasets.

The authors then walk us over the different ways of sparsifying a neural network - providing a good summary of the current state of the art.

The paper continues with the introduction of the dataset, and providing experiments that author use to support their claims.

**Summary Of The Review:**

Please see above.

---

> ### Author Response · Authors · 2022-11-15
> **Response to Reviewer JUeu**
>
> We sincerely appreciate your positive ranking overall and detailed comments, which are very encouraging to us.
>
> **Comment 1: The authors promise to open-source their dataset, however no more details are provided. Additionally one would expect a well organized list of the "10 open source repositories" used for creating the benchmark. The open-sourcing and hopefully stream-lining future benchmarks is the most valuable part of the authors' work.**
>
> * We fully agree with you on this point and we indeed have provided a well-organized list of all the "open source repositories", training configurations, and hyperparameters in Appendix A due to the space limitation. And we are currently working on preparing a set of stream-lining open-source code and will absolutely release it after ICLR.
>
> **Comment 2: I would also expect the authors to evaluate the networks on the "weak"/"easy" datasets - to confirm that their implementations/runs are consistent with what was claimed in the intro."**
>
> * Thank you for your suggestions. As you suggested, we added an evaluation of our implementations on widely used evaluation - ResNet-20 on CIFAR-10. The following table shows that all methods perform satisfactorily as we expected. We can easily find matching subnetworks at sparsity as high as $>$ 80% and dynamic sparse training (RigL) consistently brings benefits to sparse training. In stark contrast to the performance on SMC-Bench, the performance loss is rather marginal (6% vs. 74%).
> | Sparsity  | 0.2 | 0.36 | 0.48 | 0.59 | 0.67 | 0.73 | 0.79 | 0.83 | 0.86 | 0.89 |
> | ------------- | :-----------: |:-----------:|:-----------: |:-----------: |:-----------: |:-----------:|:-----------: |:-----------: |:-----------: |:-----------:|
> | Dense | 92.43 | 92.43 | 92.43 | 92.43 | 92.43 | 92.43 | 92.43 | 92.43 | 92.43 | 92.43|
> | LTH  | **92.36** | 92.28 | **92.50** | **92.21**| **92.65**|  **92.11**| **92.27**| **92.11**| **91.73**| **91.14**|
> | GMP |92.19|92.01|91.99| 91.44| 91.78| 91.44| 91.58| 91.35| 91.02| 91.09|
> | OMP (After) |92.36| **92.41**| 92.29| 92.16| 91.92| 92.03|91.43| 91.08| 90.97| 90.33|
> | SNIP + RigL | 92.03|92.10| 91.87| 91.95| 91.52| 91.24| 91.12| 90.46| 90.20| 89.16|
> | SNIP | 92.15| 91.97| 92.20| 91.47| 91.00| 90.67| 90.56|89.60| 88.84| 88.02 |

---

### Official Review · Reviewer_11zQ · 2022-10-31

**Confidence:** 5
**Clarity, Quality, Novelty And Reproducibility:** The paper is clear, original, and rep…
**Correctness:** 3
**Technical Novelty And Significance:** 1
**Empirical Novelty And Significance:** 3
**Recommendation:** 6

**Strength And Weaknesses:**

Strength
- This paper provides a range of new and difficult evaluation tasks for pruning and serves as a new benchmark.
- Overall the paper is well written.

Weaknesses
- The results achieved do not seem to provide significantly new or unknown information in the community. They basically show that iterative magnitude based pruning after training works well. The discovery is where pruning methods start to fail, but for more complex and difficult tasks it is quite obvious and expected that they will fail earlier. In this sense, the paper quite exaggerates the benefit of this new benchmark.
- This paper serves to provide empirical results but also draw comparisons and conclusions based only upon it rather than any analysis of methods themselves; it is like A is better than B because of the performance, but without providing why. Perhaps it suits the purpose of creating a benchmark though, but to point out a limitation.
- All evaluated methods are saliency-based pruning methods, among which all methods are magnitude based approaches except for random and SNIP. This is definitely a degrading factor, considering that throughout the paper the authors address in words a lot of different pruning approaches even including variational approaches and sparsity-inducing regularization based approaches.
- The benchmark is missing one critical aspect of evaluating sparsity methods: lack of consideration of the total computations needed to achieve the trained performant sparse networks. Precisely, given a network model, how many forward/backward passess (or gradient computations) are needed for the entire process of finding and training the sparse network result, for each and every method being compared? Why is this important? It is because the benchmark claims to serve as a new standard evaluation protocol providing comparisons that one is better than the other. With different budgets or without even mentioning it properly is simply not fair for that matter. Potential evaluation protocols may include (1) for a fixed budget of computations, how good one method compares to the others in terms of final accuracy or sparsity level, or (2) for a fixed target accuracy/sparsity, how efficiently/faster one method achieves the goal compared to the others. It would also be better to include, at least in words level, how simple or difficult it is to tune the algorithms or hyperparameters (which includes the winning process in the LTH). Without these considerations, this benchmark may only provide a partial result.


**Summary Of The Paper:**

This paper provides a new benchmark and results for evaluating sparsity methods on diverse tasks called SMC-Bench.
Throughout difficult tasks the paper finds that existing works on sparsity may fail at very early in the level of sparsity in comparison to what is known previously in the literature for rather simpler tasks such as imagenet classification or language modeling tasks.
This result can imply a few different things including ineffectiveness of existing methods and the need for new ones or even the need of re-thinking the benefit of sparsity itself.


**Summary Of The Review:**

This paper develops a new benchmark for evaluating pruning. There still remain concerns to be resolved for the paper to serve as a new pointer for sparsity research.

---

> ### Author Response · Authors · 2022-11-15
> **Response to Reviewer 11zQ (3/3)**
>
> **Comment 4: The benchmark is missing one critical aspect of evaluating sparsity methods: lack of consideration of the total computations needed to achieve the trained performant sparse networks. Precisely, given a network model, how many forward/backward passess (or gradient computations) are needed for the entire process of finding and training the sparse network result, for each and every method being compared? Why is this important? It is because the benchmark claims to serve as a new standard evaluation protocol providing comparisons that one is better than the other. With different budgets or without even mentioning it properly is simply not fair for that matter. Potential evaluation protocols may include (1) for a fixed budget of computations, how good one method compares to the others in terms of final accuracy or sparsity level, or (2) for a fixed target accuracy/sparsity, how efficiently/faster one method achieves the goal compared to the others. It would also be better to include, at least in words level, how simple or difficult it is to tune the algorithms or hyperparameters (which includes the winning process in the LTH). Without these considerations, this benchmark may only provide a partial result.**
>
> * Thank you for such a constructive comment! There are two types of benchmarks existing when it comes to sparsity/efficiency: (1) efficient training benchmarks like MLPerf Training [8] as well as most sparse training papers (RigL, SET, and GMP, etc); (2) efficient inference benchmarks like many post-training pruning approaches (IMP, OMP, LTH) as well as Frankle et al. (2021) [9], which compares several pruning at initialization (PaI) methods with two after-training approaches: Magnitude pruning after training and Lottery ticket rewinding (although their main goal is to assess the efficacy of these PaI methods). These two types of benchmarks are in general orthogonal and our SMC benchmark belongs to the second category. The protocol in the efficient inference benchmarks typically compares the performance of different sparse algorithms at the same sparsity, while not explicitly comparing training costs. Since the goal of SMC-Bench is to provide a more challenging benchmark beyond the capacity of the current sparse neural networks, we care more about the best performance of pruning methods at the same sparsity, rather than their training efficiency. Therefore we believe our benchmark is already self-contained.
>
> * However, we do agree that it is vital to set a fixed budget of computations when the goal is to make an apple-to-apple comparison among different pruning and sparse training methods. To respect your opinion, we added a short paragraph to discuss this in the limitation section: The main goal of our benchmark is not to provide an apple-to-apple comparison among different sparse algorithms in terms of training efficiency, but to highlight the limitations of the commonly-used evaluation protocols and assemble a new challenging one. Hence, the training costs of different sparsification approaches evaluated in this paper vary a lot from prior-training to post-training. Practical designs would need to consider the training efficiency as well, but it is out of the scope of our current work.
>
> * For the suggestions about adding discussions about difficulties to tune different algorithms and their hyperparameters, we highly appreciate it and we will add a detailed discussion in our revision.
>
> [1] Frankle, Jonathan, and Michael Carbin. "The lottery ticket hypothesis: Finding sparse, trainable neural networks." ICLR 2019.
>
> [2] Calibrate and Prune: Improving Reliability of Lottery Tickets Through Prediction Calibration. AAAI 2021.
>
> [3] Diffenderfer, James, et al. "A winning hand: Compressing deep networks can improve out-of-distribution robustness." NeurIPS 2021.
>
> [4] Liu, Shiwei, et al. "Deep ensembling with no overhead for either training or testing: The all-round blessings of dynamic sparsity." NeurIPS 2021.
>
> [5] Guo, Yiwen, et al. "Sparse dnns with improved adversarial robustness." Advances in neural information processing systems 31 (2018).
>
> [6] Chen, T., Zhang, Z., Wu, J., Huang, R., Liu, S., Chang, S., Wang, Z. Can You Win Everything with A Lottery Ticket?. TMLR.
>
> [7] Kurtic, Eldar, et al. "The Optimal BERT Surgeon: Scalable and Accurate Second-Order Pruning for Large Language Models." arXiv preprint arXiv:2203.07259 (2022).
>
> [8] Mattson, Peter, et al. "Mlperf training benchmark." Proceedings of Machine Learning and Systems 2 (2020): 336-349.
>
> [9] Frankle, Jonathan, et al. "Pruning neural networks at initialization: Why are we missing the mark?." arXiv preprint arXiv:2009.08576 (2020).

---

> > ### Comment · Reviewer_11zQ · 2022-11-26
> > **response to comment 4**
> >
> > - Thank you for adding a section to address the mentioned limitation.

---

> > > ### Author Response · Authors · 2022-11-28
> > > **Thanks to Reviewer 11zQ**
> > >
> > > Thank you for many thoughtful comments, too!
> > >
> > > We hope our follow-up answers have further clarified some of your remaining concerns. Please don't hesitate to let us know if anything more needs to be addressed.
> > >
> > > This is a piece of work that we're confident in its contribution to the sparse NN community, and we will very carefully integrate all your suggestions to improve its quality.

---

> ### Author Response · Authors · 2022-11-15
> **Response to Reviewer 11zQ (2/3)**
>
> **Comment 2:  This paper serves to provide empirical results but also draw comparisons and conclusions based only upon it rather than any analysis of methods themselves; it is like A is better than B because of the performance, but without providing why. Perhaps it suits the purpose of creating a benchmark though, but to point out a limitation.**
>
> * Thanks. Our macro goal is indeed to provide a more challenging benchmark as you also agree, not to conduct a comprehensive comparison between different sparsification methods, even though we have evaluated them with the purpose to prove our claim.
>
> * Besides, even though not our main goal, we actually have provided several detailed analyses in our submission, such as (1) the effect of not pruning the embedding layers; (2) if these are trivial layer collapse occurs across different methods; (3) visualization of the layer-wise sparsity. We have also observed several interesting findings: (1) the pruning of embedding layers is not the root cause for the failings; (2) SNIP underperforms other methods as it significantly prunes pre-trained embedding layers; (3) magnitude-based methods share an extremely similar sparsity pattern albeit their dissimilar performance. Still, we do not want to name a fixed cause for why these methods struggle to perform the tasks from SMC-Bench due to the problem's inherent complicacy, but would rather motivate this open problem for the community to study by our benchmark.
>
> **Comment 3: All evaluated methods are saliency-based pruning methods, among which all methods are magnitude based approaches except for random and SNIP. This is definitely a degrading factor, considering that throughout the paper the authors address in words a lot of different pruning approaches even including variational approaches and sparsity-inducing regularization based approaches.**
>
> * We think you have made a fair point to help improve the quality of our paper. We might humbly suggest to consider stronger second-order pruning methods than the suggested ones, since it has been shown that variational approaches and sparsity-inducing regularization-based approaches fail to perform on large-scale tasks, and the simple magnitude pruning (GMP) achieves better performance. Here we added one recently proposed SOTA second-order pruning method Optimal BeRT Surgeon (oBERT) [7] due to its promising expertise on BERT pruning (our models on Complex Reasoning tasks are also BERT based - RoBERTa). We evaluated on the Roberta base model instead of large due to the OOM issue. The hyperparameters were chosen as the follows: we adjust the num_grads as the total number of iterations in one epoch (1264 for Winogrande and 600 for CSQA), Fisher_block_size is set as B=50, and lamda is 10-7. The results are reported in below Table. Due to the memory issue, we test it with RoBERTa-based instead of RoBERTa-large which is used in our submission.  We can see that while oBERT seems is a better pruning method than LTH and GMP, it also suffers from a dramatic performance loss.
> * **RoBERTa-base on commonsense QA**:
> | Sparsity  | 0.2 | 0.36 | 0.48 | 0.59 | 0.67 | 0.73 | 0.79 | 0.83 | 0.86 | 0.89 |
> | ------------- | :-----------: |:-----------:|:-----------: |:-----------: |:-----------: |:-----------:|:-----------: |:-----------: |:-----------: |:-----------:|
> | Dense | 66.91 | 66.91 | 66.91 | 66.91 | 66.91 | 66.91 | 66.91 | 66.91 | 66.91 | 66.91 |
> | LTH  | **66.91**|  64.94|  59.37|  32.84| 19.57|  19.57|  19.57|  19.57|  **19.57**| 19.57|
> | GMP |64.78| 56.76| 43.49| 19.57| 19.57| 19.57| 19.57| 19.57| **19.57**| 19.57|
> |oBERT | 66.50| **65.11**| **62.00**| **52.91**| **37.10** | **25.88**| **22.19**| **19.90**| 16.54| **20.07**|
> * **RoBERTa-base on WinoGrande**:
> | Sparsity  | 0.2 | 0.36 | 0.48 | 0.59 | 0.67 | 0.73 | 0.79 | 0.83 | 0.86 | 0.89 |
> | ------------- | :-----------: |:-----------:|:-----------: |:-----------: |:-----------: |:-----------:|:-----------: |:-----------: |:-----------: |:-----------:|
> | Dense | 68.09 | 68.09| 68.09| 68.09| 68.09| 68.09| 68.09| 68.09| 68.09| 68.09|
> | LTH  | **67.17** | **65.72** | 61.99 | 56.55 |**54.75** | **51.93** | **51.77** | **51.65** |51.00 | 50.22|
> | GMP |67.02|65.27| **62.14**|56.78|52.82| 50.33| 50.61| 51.07| **51.98**|49.64|
> |oBERT | 66.64|**65.72**|61.38| **57.24**| 52.51| 51.06| 50.33|50.55| 50.46| **50.45**|

---

> > ### Comment · Reviewer_11zQ · 2022-11-26
> > **response to comments 2**
> >
> > - Thank you for providing reference.. but again this isn't for addressing my Comment 2; rather it appears to be repeating one of the contributions in your work (which I don't disagree with). Comment 2 is about pointing out a limitation (perhaps this is beyond the scope of the purpose of your work though, which I am not saying is anything wrong).

---

> > > ### Author Response · Authors · 2022-11-27
> > > **Thanks**
> > >
> > > Thank you for your understanding. We are also very eager to figure out the root cause for the failure of SNNs on SMC-Bench. While this paper is mainly about "posing a question", we believe it would be more scientific and rigorous to make a thorough investigation, as a future work of "finding its answer".

---

> > ### Comment · Reviewer_11zQ · 2022-11-26
> > **response to comment 3**
> >
> > - Addition of oBERT is definitely a plus especially during the short rebuttal period.
> > - I disagree with (1) "regulariazation-based approaches fail to perform on large-scale tasks" or (2) the logic that "since it has been shown that variational approaches and sparsity-inducing regularization-based approaches.." (implying) it suffices to evaluate on a second order based approach. This is neither correct or fair argument. I would suggest the authors correct the coverage of what's being evaluated in this work rather than incorrectly or unfairly describing other works based on anything unproved.

---

> > > ### Author Response · Authors · 2022-11-27
> > > **Follow-up**
> > >
> > > We appreciate your correction and constructive suggestions! We first explain that the conclusion about variational approaches and sparsity-inducing regularization-based approaches is obtained from Gale et al (https://arxiv.org/pdf/1902.09574.pdf), which states that ''While variational dropout and l0 regularization achieve state-of-the-art results on small datasets, we show that they perform inconsistently for largescale tasks and that simple magnitude pruning can achieve comparable or better results for a reduced computational budget.'' We do agree that our logic might sound improper and misleading when this context is not appropriately explained and will make sure to clarify so during the paper revision.
> > >
> > > SMC indeed shows sparse training is not perfect on larger-scale tasks and provides more performance room to motivate new methods. Also, we do NOT claim what sparse NN methods will work or not on larger-scale tasks. We focus on testing saliency-based pruning approaches (LTH, GMP, OMP, etc) since they are currently the most popular and effective approaches on existing sparse benchmarks. While we observe that variational dropout and l0 regularization seem to not work well on existing benchmarks (https://arxiv.org/pdf/1902.09574.pdf), we also don’t exclude the possibility that they can be tuned to be more effective on larger SMC. We leave it as future work and emphasize again our goal is NOT to test every possible sparse training algorithm in one paper, or to invent any new one.

---

> ### Author Response · Authors · 2022-11-15
> **Response to Reviewer 11zQ (1/3)**
>
> We thank you for your time and for providing very constructive comments. We provide point-wise responses to your comments below.
>
> **Comment 1: The results achieved do not seem to provide significant new or unknown information in the community. They basically show that iterative magnitude-based pruning after training works well. The discovery is where pruning methods start to fail, but for more complex and difficult tasks it is quite obvious and expected that they will fail earlier. In this sense, the paper quite exaggerates the benefit of this new benchmark.**
>
> * We respectfully yet firmly disagree with you. We believe with high confidence that our findings and contributions are new, significant, and valuable for the sparsity community. We also invite you to review the unanimously positive appreciation of our novelty and significance from the other three reviewers: **Reviewer JUeu**: "an obvious question that seems not to have been addressed yet..."; **Reviewer wzLQ**: "a very interesting new angle'', "offer brand-new knowledge of broad interest to the sparse NN community'', "correctly remark", "Novelty: high", "exciting"; **Reviewer yFvQ**: "it will make a big/good splash in the sparsity community", "High technical quality and novelty as a "new perspective" benchmark in sparsity and pruning", etc.
>
> * First of all, we would like to clarify that our overarching research goal is not to discover where pruning methods start to fail, but instead to alarmingly highlight the limitations of the commonly-used **evaluation standards (not sparsification algorithms)** that are overlooked by the sparsity community: SNNs are narrowly evaluated only with a single or sometimes two tasks on well-understood datasets on which SNNs have already proven their proficiency. Such restricted and simple evaluations will limit the development of sparse neural networks and are ill-suited to identify new and unexpected capabilities of SNNs. To prove our claim as well as encourage the development of highly generalizable sparse algorithms, we assemble the SMC-Bench benchmark on which all of the SOTA sparse algorithms bluntly fail to perform.
>
> * Secondly, we also disagree that it is "quite obvious" that sparse NNs will fail earlier on more different tasks. To the best of our knowledge, no prior works have ever demonstrated this phenomenon - and we hope you would agree too that nothing in science was obvious before it was discovered. Indeed, a large body of works have shown that sparse NNs seem can easily win over the dense NNs on various tasks, including performance [1], uncertainty estimation [2], out-of-distribution generalization [3,4], adversarial robustness [5], and even a win of everything [6]. We for the first time show that all sparse NNs fail to perform more complex tasks that require reasoning and structural information - that is a strong message deserving to be passed on.

---

> > ### Comment · Reviewer_11zQ · 2022-11-26
> > **response to comment 1**
> >
> > I'm sorry for my late response.. I thank and respect your addressing. But my point (Comment 1) remains not resolved yet.
> >
> > - I view your first bullet point to Comment 1 (whether "the discovery" is significant or not) is a defense strategy rather than directly addressing my initial point as you're simply quoting other reviewers' comment.
> > - Yes, I agree that your evaluation on a wide variety of different tasks is definitely a good contribution. But that wasn't the point of my Comment 1.
> > - Your comment "nothing in science was obvious before it was discovered" is agreeable, but that's not provide sufficiency. More importantly, your argument on "NN failing earlier on more *difficult* (fixed) tasks is not obvious" is very unintuitive to me. This can be understood from many different angles including the pruning tasks itself! Performance degrading as per increasing sparsity (in any standard evaluation for pruning) indicates that with smaller number of parameter budget is harder to fit the given data for the same task in representation learning; this in other words means for a more difficult representation learning task it is _expected_ to see less performing for same parameter budget. Your "no prior works have ever demonstrated" is once again, too exaggerated, which can be highly misleading.

---

> > > ### Author Response · Authors · 2022-11-27
> > > **Thank you for your response**
> > >
> > > We agree with you that it is intuitively true that sparse NNs might fail earlier on more complex tasks. However, prior works show sparse NNs generally succeed almost universally, regardless of models, tasks, and datasets. We conjecture the "always work" observation can be misleading because the standard evaluation (tasks and datasets) for pruning/sparse training is too simple, which may significantly impede future progress if left undiscovered. In order to further push the development of highly generalizable sparse algorithms, we assemble SMC-Bench which is more challenging for current SNNs to perform. Given the enormous that sparse neural networks have achieved, we believe our paper holds its merits to the community, in terms of encouraging people to rethink the prevailing win of sparsity, as well as assisting in building next-generation sparse algorithms with better potentials to generalize on complex and practical tasks.

---

> ### Author Response · Authors · 2022-11-22
> **We are keen to discuss further with you**
>
> Dear Reviewer **11zQ**,
>
> We thank reviewer **11zQ** for the time of reviewing and the constructive comments again. We really hope to discuss further with you to see if our response solves your concerns.
>
> In our response, we have (1) re-iterated our research goal and contributions to the community; (2) added a second-order pruning approach (oBERT).
>
> We genuinely hope reviewer **11zQ** could kindly check our response. Thank you!
>
> Best wishes,
>
> Authors

---

> ### Author Response · Authors · 2022-11-26
> **We are keen to discuss further with you**
>
> Dear Reviewer **11zQ**,
>
> We thank you again for your time and your constructive comments.
>
> As the discussion period is approaching its end, we would really appreciate it if you could kindly let us know whether there are any further questions. We will be more than happy to address them.
>
> Best wishes,
>
> Authors

---

### Author Response · Authors · 2022-11-18
**Clarification of our oBERT evaluation**

In the spirit of academic fairness, we clarify that the evaluation of oBERT [1] does not strictly follow the full training recipe used in the original paper. Instead, we directly plug the second-order pruning criterion of oBERT in SMC-Bench for a fair comparison between oBERT and the other pruning methods, using the default configurations of RoBERTa in Fairseq. However, we do not exclude the possibility that the promising training recipe used in the original oBERT (e.g., knowledge distillation, longer fine-tuning time, cyclic learning rate with learning rate rewinding) can help to perform on SMC-Bench, while they might as well help other pruning methods tested in the paper. Due to the limited rebuttal time window, we would like to defer a thorough investigation to future work and we will report later here.

[1] Kurtic, Eldar, et al. "The Optimal BERT Surgeon: Scalable and Accurate Second-Order Pruning for Large Language Models." arXiv preprint arXiv:2203.07259 (2022).

---

### Decision · Program_Chairs · 2023-01-20

**Decision:**

Accept: notable-top-25%

**Justification For Why Not Higher Score:**

The results are impactful but not enough to warrant the "Accept with Oral" category. It is hard to predict whether the paper will impact the field going forward in a very significant manner, as there are some concerns such as lack of computational time as metric or exclusion of certain methods.

**Justification For Why Not Lower Score:**

The paper is likely to have a sizeable impact on the field by focusing it on more challenging tasks.

**Metareview: Summary, Strengths And Weaknesses:**

The paper presents a new benchmark for evaluating the performance of sparse neural networks. The authors argue that existing benchmarks are too simple and do not reflect (all) real use cases of sparsity. To address this issue, the authors introduce the "Sparsity May Cry" Benchmark (SMC-Bench).

The authors evaluate a range of state-of-the-art sparse algorithms on this benchmark, and find that many of them perform poorly, even at low levels of sparsity. This observation highlights the need for the development of more generic sparse algorithms.

One reviewer raised the important point that the benchmark does not adequately consider the total computational cost of training a sparse network. This hasn't been addressed by the Authors. I agree it is a key point, but at the same time the benchmark is sufficiently challenging to help drive the field even with this limitation.

Another important weakness is lack of breadth in evaluated algorithms. Authors have partially addressed this during rebuttal. I believe Authors have shown sufficient evidence that the benchmark is challenging, and future work can evaluate more diverse methods.

The key strength of the paper is helping define a new challenge for the field, which will likely help drive future progress.

Overall, the paper presents a valuable contribution to the field by introducing a new, challenging benchmark for evaluating sparse neural networks. Its acceptance (ultimately the least supportive reviewer also leaned towards accepting the paper) by all reviewers highlights the importance of this work and the need for further research in this area.

Thank you for your submission. Please remember to address all reviewers' comments in the final camera ready version.

**Note From Pc:**

if the above contains the word "oral" or "spotlight" please see: "oral" presentation means -> notable-top-5% and "spotlight" means -> notable-top-25%. As stated in our emails, we are disassociating presentation type from AC recommendations